# Simple and Fast Algorithm for Binary Integer and Online Linear Programming

**Xiaocheng Li**[†]     **Chunlin Sun**[‡]     **Yinyu Ye**[†]

[†]Department of Management Science and Engineering, Stanford University
[‡] Institute for Computational and Mathematical Engineering, Stanford University
{chengli1, chunlin, yinyu-ye}@stanford.edu

## Abstract

In this paper, we develop a simple and fast online algorithm for solving a class of binary integer linear programs (LPs) arisen in general resource allocation problem. The algorithm requires only one single pass through the input data and is free of doing any matrix inversion. It can be viewed as both an approximate algorithm for solving binary integer LPs and a fast algorithm for solving online LP problems. The algorithm is inspired by an equivalent form of the dual problem of the relaxed LP and it essentially performs (one-pass) projected stochastic subgradient descent in the dual space. We analyze the algorithm under two different models, stochastic input and random permutation, with minimal technical assumptions on the input data. The algorithm achieves $O\left(m\sqrt{n}\right)$ expected regret under the stochastic input model and $O\left((m + \log n)\sqrt{n}\right)$ expected regret under the random permutation model, and it achieves $O(m\sqrt{n})$ expected constraint violation under both models, where $n$ is the number of decision variables and $m$ is the number of constraints. In addition, we employ the notion of permutational Rademacher complexity and derive regret bounds for two earlier online LP algorithms for comparison. Both algorithms improve the regret bound with a factor of $\sqrt{m}$ by paying more computational cost. Furthermore, we demonstrate how to convert the possibly infeasible solution to a feasible one through a randomized procedure. Numerical experiments illustrate the general applicability and effectiveness of the algorithms.

## 1   Introduction

In this paper, we present a simple and fast online algorithm to approximately solve a general class of binary (integer) linear programs (LP). Different specifications of the considered LP problem cover a wide range of classic problems and modern applications: secretary problem [13], knapsack problem [17], resource allocation problem [29], generalized assignment problem [8], network routing problem [5], matching problem [23], etc. From the perspective of **integer LP**, our algorithm is an efficient approximate algorithm that features for provable performance guarantee. In general, integer LP is NP-complete and the LP relaxation technique is widely used in designing integer LP algorithm. Our algorithm is inspired by the relaxed LP, and it outputs an integer solution to the relaxed LP (so that there is no need for a rounding procedure). The solution can thus be viewed as an approximate solution to both the integer LP and the relaxed LP. From the perspective of **online LP**, to the best of our knowledge, our algorithm is the most simple and efficient online LP algorithm so far. Furthermore, the algorithm analysis is conducted under the two prevalent models: stochastic input model and random permutation model. The stochastic input model assumes that the columns of the LP together with the corresponding coefficients in the objective function are drawn i.i.d. from an unknown distribution.

The contribution of this paper can be summarized in the following three aspects:

- We develop a fast online algorithm to solve a general class of LPs. The algorithm identifies an equivalent form of the dual problem and performs projected stochastic subgradient descent to solve the dual problem. The online LP algorithms discussed in literature [3, 18, 24, 15, 20] all involve solving scaled LPs throughout the process but our algorithm is free of solving any linear programs or matrix inversions. Specifically, our algorithm has an $O(\text{nnz}(\boldsymbol{A}))$ flop complexity (linear in the number of non-zero entries in $\boldsymbol{A}$), while the previous OLP algorithms all require solving $O(\log n)$ or $O(n)$ of LPs (increasing to the full size over time). For example, [3] solved $O(\log n)$ LPs and [18] solved $O(n)$ LPs. As far as we know, the algorithm is the first of its kind and the most efficient OLP algorithm so far.

- We derive upper bounds for the regret and constraint violation of our algorithm with minimal statistical assumptions. Under the stochastic input model (Section 3), we only assume the boundedness of the LP entries. The assumption is weaker than [20] because we do not require a strong convexity for the underlying stochastic program, and it is weaker than [21] for that we do not assume a finite support of the random coefficients. Under the random permutation model (Section 4), our assumption is also weaker than all previous works [3, 18, 24, 15] in that we allow negative data values for the input of the LP. The relaxation of the non-negativeness assumption entails a entirely different approach to analyze the regret/competitiveness ratio. Specifically, our analysis utilizes both the structure of the problem and the property of gradient-based algorithm, and incorporates concentration arguments under both stochastic input and random permutation models.

- We perform more algorithm analyses under the random permutation model through the notion of Permutational Rademacher Complexity [27] which is previously designed for the analysis of transductive learning. We show how the notion can be used for analyzing online LP algorithms or possibly more general online algorithms under the random permutation model. The results show that two previous algorithms [3, 18] improves the regret bound with a factor of $\sqrt{m}$ by paying more computational cost.

The algorithms developed in this paper can be viewed as a stochastic algorithm to solve large-scale (integer) LPs. The literature on large-scale LP algorithms traced back to the early works on column generation algorithm [14, 9]. In recent years, statistical structures underlying the input of LP have been taken into consideration. Sampling-based/randomized LP algorithms are derived to handle large number of constraints in the LP of Markov Decision Processes [10, 19], the standard form of LP [30], robust convex optimization [6], etc. Compared to this line of works, our algorithms utilize the dual LP and are free of solving any small-scale or reduced-size LP. Our algorithm can also be viewed as an online and efficient version of the dual projected subgradient (DPG) algorithm for LP [4]. Our algorithm employs one column for subgradient descent in each iteration, whereas the dual project subgradient algorithm requires the whole constraint matrix and conducts matrix multiplication in each iteration. In addition, a class of backpressure/max-weight algorithms [25] are developed in the control/queueing literature and the backpressure algorithm can be interpreted from a view of pressure gradient. The key distinction between the backpressure algorithm and our algorithm lies in the objective function: the backpressure algorithm aims to ensure the stability of a queueing networks while our algorithm aims to maximize the revenue/reward obtained thoughout an online procedure.

Our work also complements to the literature of online convex optimization with constraints (`OCOwC`) [22, 31, 32]. The key difference between the online LP problem and the `OCOwC` problem is that when computing the regret, the former considers a dynamic oracle where the decision variables are allowed to take different values at each time period, while the later considers a stationary benchmark where the the decision variables are required to be the same at each time period.

## 2 Integer Linear Program and Main Algorithm

### 2.1 Integer LP, Primal LP, and Dual LP

Consider the binary integer LP

$$
\begin{aligned}
\max\ \ & \boldsymbol{r}^\top \boldsymbol{x} \\
\text{s.t.}\ \ & \boldsymbol{A}\boldsymbol{x} \le \boldsymbol{b} \\
& x_j \in \{0,1\},\ \ j = 1, ..., n
\end{aligned} \tag{1}
$$

where $\boldsymbol{r} = (r_1, ..., r_n)^\top \in \mathbb{R}^n$, $\boldsymbol{A} = (\boldsymbol{a}_1, ..., \boldsymbol{a}_n) \in \mathbb{R}^{m \times n}$, and $\boldsymbol{b} = (b_1, ..., b_m)^\top \in \mathbb{R}^m$. Here $\boldsymbol{a}_j = (a_{1j}, ..., a_{mj})^\top$ denotes the $j$-th column of the constraint matrix $\boldsymbol{A}$. The decision variables $\boldsymbol{x} = (x_1, ..., x_n)^\top$ are binary integers. An LP relaxation of the above problem is

$$\max \ \boldsymbol{r}^\top \boldsymbol{x} \tag{2}$$
$$\text{s.t. } \boldsymbol{Ax} \leq \boldsymbol{b}$$
$$\boldsymbol{0} \leq \boldsymbol{x} \leq \boldsymbol{1}.$$

The dual problem of (2) is

$$\min \ \boldsymbol{b}^\top \boldsymbol{p} + \boldsymbol{1}^\top \boldsymbol{s} \tag{3}$$
$$\text{s.t. } \boldsymbol{A}^\top \boldsymbol{p} + \boldsymbol{s} \geq \boldsymbol{r}$$
$$\boldsymbol{p} \geq \boldsymbol{0}, \boldsymbol{s} \geq \boldsymbol{0},$$

where the decision variables are $\boldsymbol{p} \in \mathbb{R}^m$ and $\boldsymbol{s} \in \mathbb{R}^n$. Throughout this paper, $\boldsymbol{0}$ and $\boldsymbol{1}$ denote all-zero and all-one vector, respectively. We will use ILP (1), P-LP (2), and D-LP (3) to refer to both the optimization problems and their optimal objective values. Evidently, we have the follow relation between the optimal objective values,

$$\text{ILP}(1) \leq \text{P-LP}(2) = \text{D-LP}(3).$$

This relation lays the foundation for the wide application of LP relaxation in solving integer linear programs [8]. Denote the optimal solutions to (2) and (3) with $\boldsymbol{x}^*$, $\boldsymbol{p}_n^*$, and $\boldsymbol{s}^*$, and the optimal solutions to (1) as $\bar{\boldsymbol{x}}^*$. From the complementary slackness condition, we know that

$$x_j^* = \begin{cases} 1, & r_i > \boldsymbol{a}_j^\top \boldsymbol{p}_n^* \\ 0, & r_i < \boldsymbol{a}_j^\top \boldsymbol{p}_n^* \end{cases} \tag{4}$$

for $j = 1, ..., n$. When $r_j = \boldsymbol{a}_j^\top \boldsymbol{p}_n^*$, the optimal solution $x_j^*$ may take a non-integer value. The implication of this optimality condition is that the primal optimal solution $\boldsymbol{x}^*$ can be largely determined by the dual optimal solution $\boldsymbol{p}_n^*$.

## 2.2 Main Algorithm

The derivation of our algorithm relies on the following observation. If we denote the right-hand-side $\boldsymbol{b} = n\boldsymbol{d}$, an equivalent form of the dual problem that only involves decision variables $\boldsymbol{p}$ can be obtained from (3) by plugging the constraints into the objective and removing the dual decision variables $\boldsymbol{s}$.

$$\min_{\boldsymbol{p} \geq \boldsymbol{0}} f_n(\boldsymbol{p}) = \boldsymbol{d}^\top \boldsymbol{p} + \frac{1}{n} \sum_{j=1}^n \left( r_j - \boldsymbol{a}_j^\top \boldsymbol{p} \right)^+ \tag{5}$$

where $(\cdot)^+$ denotes the positive part function.

Now, we present the main algorithm – Simple Online Algorithm. It is a dual-based online algorithm that observes the inputs of the LP sequentially and decides the value of decision variable $x_t$ immediately after each observation $(r_t, \boldsymbol{a}_t)$. At each time $t$, it updates the vector with the new observation $(r_t, \boldsymbol{a}_t)$ and projects to the non-negative orthant to ensure the dual feasibility.

The key of the algorithm is the updating formula for $\boldsymbol{p}_t$, namely Step 5 in Algorithm 1. For two vectors $\boldsymbol{u}, \boldsymbol{v} \in \mathbb{R}^m$, $\boldsymbol{u} \vee \boldsymbol{v} = (\max\{u_1, v_1\}, ..., \max\{u_m, v_m\})^\top$ denotes the elementwise maximum operator. Specifically, the update from $\boldsymbol{p}_t$ to $\boldsymbol{p}_{t+1}$ can be interpreted as a *projected stochastic subgradient descent* method for optimizing the problem (5). Concretely, the subgradient of the $t$-th term in (5) evaluated at $\boldsymbol{p}_t$ is as follows,

$$\partial_{\boldsymbol{p}} \left( \boldsymbol{d}^\top \boldsymbol{p} + \left( r_t - \boldsymbol{a}_t^\top \boldsymbol{p} \right)^+ \right) \Big|_{\boldsymbol{p}=\boldsymbol{p}_t} = \boldsymbol{d} - \boldsymbol{a}_t I(r_t > \boldsymbol{a}_t^\top \boldsymbol{p}) \Big|_{\boldsymbol{p}=\boldsymbol{p}_t} = \boldsymbol{d} - \boldsymbol{a}_t x_t$$

where the second line is due to the specification of $x_t$ as the step 4 in the Algorithm 1. Throughout this paper, $I(\cdot)$ denotes the indicator function. The dual updating rule indeed implements the stochastic subgradient descent in the dual space. We defer the rigorous analyses of the algorithm performance and the choice of the step size $\gamma_t$ to later sections. As for the computational aspect, Algorithm 1 requires only one pass through the data and is free of matrix multiplications.

---

**Algorithm 1** Simple Online Algorithm

---
1: Input: $\boldsymbol{d} = \boldsymbol{b}/n$
2: Initialize $\boldsymbol{p}_1 = \boldsymbol{0}$
3: **for** $t = 1, ..., n$ **do**
4:     Set
$$x_t = \begin{cases} 1, & r_t > \boldsymbol{a}_t^\top \boldsymbol{p}_t \\ 0, & r_t \leq \boldsymbol{a}_t^\top \boldsymbol{p}_t \end{cases}$$

5:     Compute
$$\boldsymbol{p}_{t+1} = \boldsymbol{p}_t + \gamma_t \left( \boldsymbol{a}_t x_t - \boldsymbol{d} \right)$$
$$\boldsymbol{p}_{t+1} = \boldsymbol{p}_{t+1} \vee \boldsymbol{0}$$

6: **end for**
7: Output: $\boldsymbol{x} = (x_1, ..., x_n)$

---

### 2.3 Performance Measures

We analyze the algorithm under two metrics – optimality gap/regret and constraint violation. The optimality gap measures the difference in objective values for the algorithm output and the true optimal solution. The bi-objective performance measure is widely used in the literature on the online convex optimization with constraints (OCOwC). Specifically, the same objective is considered in [22, 31, 32, 1, 2]. In the following two sections, we will formalize the assumptions and analyze the algorithm in two different settings.

## 3 Stochastic Input Model

In this section, we formalize and analyze the algorithm under the statistical assumption proposed in the last section. Concretely, we discuss the performance of Algorithm 1 when the inputs of an (integer) LP follow the stochastic input model which assumes the column-coefficient pair $(r_j, \boldsymbol{a}_j)$'s are i.i.d. generated. LPs and integer LPs that satisfy this model naturally arise from some application contexts where each pair represents a customer/order/request. In particular, at each time $t$, $\boldsymbol{a}_t$ can be interpreted as a customer request for the resources while $r_t$ represents the revenue that the decision maker receives from accepting this request. The binary decision variable $x_t$ represents the decision of acceptance or rejection of the $t$-th request.

### 3.1 Assumptions and Performance Measures

The following assumption formalizes the statistical assumption on $(r_j, \boldsymbol{a}_j)$ in an i.i.d. setting.

**Assumption 1** (Stochastic Input). *We assume*

    *(a) The column-coefficient pair $(r_j, \boldsymbol{a}_j)$'s are i.i.d. sampled from an unknown distribution $\mathcal{P}$.*

    *(b) There exist constants $\bar{r}$ and $\bar{a}$ such that $|r_j| \leq \bar{r}$ and $\|\boldsymbol{a}_j\|_\infty \leq \bar{a}$ for $j = 1, ..., n$.*

    *(c) The right-hand-side $\boldsymbol{b} = n\boldsymbol{d}$ and there exist $\underline{d}, \bar{d} \in \mathbb{R}_+$ such that $\underline{d} \leq d_i \leq \bar{d}$ for $i = 1, ..., m$.*

We emphasize that the constants $\bar{r}$, $\bar{a}$, $\underline{d}$ and $\bar{d}$ only serve for analysis purpose and are assumed unknown a priori for algorithm implementation. Also, we allow dependence between components in $(r_j, \boldsymbol{a}_j)$'s. Besides the boundedness, we have put minimal assumption on $r_j$ and $\boldsymbol{a}_j$. For part (c), the assumption on right-hand-side side provides a service level guarantee, i.e., it ensures a fixed proportional of customers/orders can be fulfilled as the total number of customers (market size) $n$ increases. We use $\Xi$ to denote the family of distributions that satisfy Assumption 1 (b).

Next, we formally define the regret and the constraint violation. Denote the optimal objective values of the ILP and P-LP as $Q_n^*$ and $R_n^*$, respectively. The objective value obtained by the algorithm output is
$$R_n = \sum_{j=1}^n r_j x_j.$$

The quantity of interest is the optimality gap $Q_n^* - R_n$, which has an upper bound $Q_n^* - R_n \leq R_n^* - R_n$. The expected optimality gap is $\Delta_n^{\mathcal{P}} = \mathbb{E}\left[R_n^* - R_n\right]$ where the expectation is taken with respect to $(r_j, \boldsymbol{a}_j)$'s. Define regret as the worst-case optimality gap

$$\Delta_n = \sup_{\mathcal{P} \in \Xi} \Delta_n^{\mathcal{P}}.$$

Thus the regret bound derived in this paper has a two-fold meaning: (i) an upper bound for the optimality gap of solving the integer LP; (ii) a regret bound for the regret of solving online LP problem. Provided that we do not assume any knowledge of the distribution $\mathcal{P}$, this type of distribution-free bound is legitimate. We emphasize that the definition of regret for the canonical online LP problem differs from that for the online convex optimization problem [16] where the decision variables for the offline optimal are restricted to take the same value over time; in contrast, here we allow $x_1, ..., x_n$ to take different values and thus consider a dynamic oracle in defining $R_n^*$. Another performance measure for Algorithm 1 is the *constraint violation*,

$$v(\boldsymbol{x}) = \left\| \left( \boldsymbol{A}\boldsymbol{x} - \boldsymbol{b} \right)^+ \right\|_2$$

where $\boldsymbol{A}$ is the constraint coefficient matrix, $\boldsymbol{b}$ is the right-hand-side constraint, and $\boldsymbol{x}$ is the solution. We aim to quantify the expected $L_2$ norm of the constraint violation. Similar to the regret, we seek for an upper bound for the constraint violation that is not dependent on the distribution $\mathcal{P}$.

## 3.2  Algorithm Analyses

First, we analyze the dual price sequence $\boldsymbol{p}_t$'s. The following lemma states that the dual price $\boldsymbol{p}_t$'s under Algorithm 1 will remain bounded throughout the process, and this is true with probability 1.

**Lemma 1.** *Under Assumption 1, if the step size $\gamma_t \leq 1$ in Algorithm 1, then $\|\boldsymbol{p}^*\|_2 \leq \frac{\bar{r}}{\underline{d}}$, and*

$$\|\boldsymbol{p}_t\|_2 \leq \frac{2\bar{r} + m(\bar{a} + \bar{d})^2}{\underline{d}} + m(\bar{a} + \bar{d}).$$

*with probability 1 for $t = 1, ..., n$, where $\boldsymbol{p}_t$'s are specified by Algorithm 1.*

Essentially, this boundedness property arises from the updating formula. The intuition is that if the dual price $\boldsymbol{p}_t$ becomes large, then most of the "buying" requests (with $\boldsymbol{a}_j$ being positive) will not be rejected, and this will lead to a decrease of the dual price when computing $\boldsymbol{p}_{t+1}$. As we will see later, the norm of $\boldsymbol{p}_t$ will appear frequently in the algorithm performance analyses. Therefore the implicit boundedness of $\boldsymbol{p}_t$ becomes important in that it saves us from doing explicit projection. On one hand, projecting $\boldsymbol{p}_t$ into a compact set at every step might be computational costly; on the other hand, this compact set requires more prior knowledge on underlying LP.

**Theorem 1.** *Under Assumption 1, if the step size $\gamma_t = \frac{1}{\sqrt{n}}$ for $t = 1, ..., n$, the regret and expected constraint violation of Algorithm 1 satisfy*

$$\mathbb{E}[R_n^* - R_n] \leq m(\bar{a} + \bar{d})^2 \sqrt{n}$$

$$\mathbb{E}\left[v(\boldsymbol{x})\right] \leq \left( \frac{2\bar{r} + m(\bar{a} + \bar{d})^2}{\underline{d}} + m(\bar{a} + \bar{d}) \right) \sqrt{n}.$$

*hold for all $m, n \in \mathbb{N}^+$ and distribution $\mathcal{P} \in \Xi$.*

The number of constraints $m$ decides the dimension of the dual price vectors $\boldsymbol{p}_t$'s. Both the regret and the expected constraint violation is $O(m\sqrt{n})$. Algorithm 1 conducts subgradient descent updates in the dual space but the performance is measured by the primal objective. The key idea for the proof of Theorem 1 is to establish the connections between primal objective, dual objective, and constraints violation through the lens of the updating formula for $\boldsymbol{p}_t$. The proof mimics the classic analysis for convex online optimization problems [16]. This provides an explanation for why the seemingly related problems of online LP and online convex optimization with constraints (`OCOwC`) are studied separately in the literature. On one hand, the online LP literature has been focused on studying the primal objective value as the performance measure. On the other hand, the `OCOwC` problem [22, 31, 32] also studied mainly the primal objective under online stochastic subgradient descent algorithms. However, it is the dual problem of online LP that corresponds to a special form of the primal problem in the `OCOwC` literature. Our contribution is to identify this correspondence and to establish the primal-dual connection for online LP problem when applying stochastic subgradient descent.

# 4 Random Permutation Model

In this section, we consider a random permutation model where the column-coefficient pair $(r_j, \boldsymbol{a}_j)$ arrives in a random order. The values of $(r_j, \boldsymbol{a}_j)$'s can be chosen adversarially at the start. However, the arrival order of $(r_j, \boldsymbol{a}_j)$'s is uniformly distributed over all the permutations. There are two ways to interpret Algorithm 1 under this random permutation model. First, it can be interpreted as an online algorithm that solves an online LP problem under data generation assumptions that are weaker than the i.i.d. assumptions discussed in the last section. The stochastic input model therefore can be viewed as a special case of the random permutation model. Second, from the perspective of solving integer LPs, the permutation creates the randomness for integer LPs when there is no inherent randomness with the coefficients. As we will see, this artificially created randomness is sufficient to provide provable performance guarantee for Algorithm 1 which is comparable to the case of the stochastic input model. In this section, we analyze the regret and the constraint violation of Algorithm 1 under the random permutation model, and later in Section 5, we provide a more systematic treatment of the random permutation model and analyze the performance of two previously proposed algorithms.

## 4.1 Assumption and Performance Measures

In parallel to the stochastic input model, we formalize the random permutation model as follows.

**Assumption 2** (Random Permutation). *We assume*

  (a) *The column-coefficient pair $(r_j, \boldsymbol{a}_j)$ arrives in a random order.*

  (b) *There exist constants $\bar{r}$ and $\bar{a}$ such that $|r_j| \leq \bar{r}$ and $\|\boldsymbol{a}_j\|_\infty \leq \bar{a}$ for $j = 1, ..., n$.*

  (c) *The right-hand-side $\boldsymbol{b} = n\boldsymbol{d}$ and there exist $\underline{d}, \bar{d} \in \mathbb{R}_+$ such that $\underline{d} \leq d_i \leq \bar{d}$ for $i = 1, ..., m$.*

Assumption 2 part (b) and (c) are identical to the stochastic input model. Denote the input data set $\mathcal{D} = \{(r_j, \boldsymbol{a}_j) : 1 \leq j \leq n\}$. Part (a) in Assumption 2 states that we observe a permuted realization of the data set. Additionally, we make the following assumption on the data set $\mathcal{D}$.

**Assumption 3.** *The problem inputs are in a general position, namely for any price vector $\boldsymbol{p}$, there can be at most $m$ columns such that $\boldsymbol{a}_j^\top \boldsymbol{p} = r_j$.*

This assumption is not necessarily true for all the data set $\mathcal{D}$. However, as pointed out by [11], one can always randomly perturb $r_t$'s by arbitrarily small amount. In this way, the assumption will be satisfied, and the effect of this perturbation on the objective can be made arbitrarily small. Define $x_j(\boldsymbol{p}) = I(r_j > \boldsymbol{a}_j^\top \boldsymbol{p})$ and $\boldsymbol{x}(\boldsymbol{p}) = (x_1(\boldsymbol{p}), ..., x_n(\boldsymbol{p}))$. Lemma 2 tells that $\boldsymbol{x}(\boldsymbol{p}_n^*)$ should be feasible and close to the primal optimal solution.

**Lemma 2** (Lemma 1 in [3]). *$x_j(\boldsymbol{p}_n^*) \leq x_j^*$ for all $j = 1, ..., n$ and under Assumption 3, $x_j(\boldsymbol{p}_n^*)$ and $x_j^*$ differs for no more than $m$ values of $j$. It implies that, under Assumption 3, if one uses the optimal dual solution $\boldsymbol{p}_n^*$ in the thresholding rule, the obtained solution will no greater than the primal optimal solution and they will be different for at most $m$ entries.*

As for the performance measure, we use the same notations as in Section 3.1. The expected optimality gap $\delta_n^\mathcal{D} = R_n^* - \mathbb{E}[R_n]$. Throughout this section, the expectation is always taken with respect to a random permutation on the data set $\mathcal{D}$, unless otherwise stated. Given the data set $\mathcal{D}$, $R_n^*$ is a deterministic quantity, so it is unnecessary to take an expectation for it. Define regret as the worst-case optimality gap

$$\delta_n = \sup_{\mathcal{D} \in \Xi_D} \delta_n^\mathcal{D}$$

where $\Xi_D$ denotes all the data sets that satisfy Assumption 2 (b) and Assumption 3. In this way, the regret quantifies the worst-case performance of the algorithm for all possible inputs data $\mathcal{D}$.

## 4.2 Algorithm Analyses

First, the following lemma states that the boundedness property of the dual price remains the same as in the stochastic input model. Its proof is identical to the stochastic input model, since the proof of Lemma 1 only relies on the boundedness assumption on $(r_j, \boldsymbol{a}_j)$'s but not the statistical assumption.

**Lemma 3.** *Under Assumption 2 and Assumption 3, if the step size $\gamma_t \leq 1$ in Algorithm 1, we have $\|\boldsymbol{p}_n^*\|_2 \leq \frac{\bar{r}}{\underline{d}}$, and*

$$\|\boldsymbol{p}_t\|_2 \leq \frac{2\bar{r} + m(\bar{a} + \bar{d})^2}{\underline{d}} + m(\bar{a} + \bar{d}).$$

*with probability 1 for all $t$, where $\boldsymbol{p}_t$'s are specified by Algorithm 1.*

To facilitate our derivation, we define a scaled version of the primal LP (2),

$$\max \quad \sum_{j=1}^{s} r_j x_j \qquad\qquad\qquad (s\text{-S-LP})$$

$$\text{s.t.} \quad \sum_{j=1}^{s} a_{ij} x_j \leq \frac{sb_i}{n}$$

$$0 \leq x_j \leq 1 \quad \text{for } j = 1, ..., s.$$

for $s = 1, ..., n$. Denote its optimal objective value as $R_s^*$.

**Proposition 1.** *For $s > \max\{16\bar{a}^2, e^{16\bar{a}^2}, e\}$, the following inequality holds*

$$\frac{1}{s}\mathbb{E}\left[R_s^*\right] \geq \frac{1}{n}R_n^* - \frac{m\bar{r}}{n} - \frac{\bar{r}\log s}{\underline{d}\sqrt{s}} - \frac{m\bar{r}}{s}. \qquad\qquad (6)$$

*for all $s \leq n \in \mathbb{N}^+$ and $\mathcal{D} \in \Xi_D$.*

Intuitively, in the random permutation model, the observations $\{(r_j, \boldsymbol{a}_j)\}_{j=1}^{s}$ collected until time $s$ are less informative to infer the future observations than the case of the stochastic input model. However, Proposition 1 tells that the scaled LP ($s$-S-LP) constructed based on the first $s$ observations will achieve a similar expected optimal objective value (after scaling) compared with the original problem with all $n$ observations. Note that $\mathbb{E}[R_s^*]/s = \mathbb{E}[R_n^*]/n$ is evidently true in the stochastic input model, where the expectation is taken with respect to the distribution $\mathcal{P}$. The additional terms on the right-hand-side of (6) captures the information toll (on the order of $\log s/\sqrt{s}$) for the assumption relaxation from the stochastic input model to the random permutation model.

**Theorem 2.** *Under Assumption 2 and 3, if the step size $\gamma_t = \frac{1}{\sqrt{n}}$ for $t = 1, ..., n$, the regret and expected constraint violation of Algorithm 1 satisfy*

$$R_n^* - \mathbb{E}[R_n] \leq O\left((m + \log n)\sqrt{n}\right)$$

$$\mathbb{E}\left[v(\boldsymbol{x})\right] \leq O(m\sqrt{n})$$

*for all $m, n \in \mathbb{N}^+$ and $\mathcal{D} \in \Xi_D$.*

Compared to the stochastic input model, the regret upper bound under random permutation model contains an extra term of $O(\sqrt{n}\log n)$, while the constraint violation in two models enjoys the same upper bound. Note that Proposition 1 and Theorem 2 do not require the non-negativeness assumption of the LP input. As far as we know, this is the first online LP analysis under the random permutation model without the non-negativeness assumption [24, 3, 18, 15].

## 5 Performance Analyses of Two "Slower" Algorithms

In this section, we analyze the regret of two "slower" algorithms [3, 18] of online LP under the random permutation model. Since they all involved solving scaled LPs, they are slower than the algorithm proposed in this paper. Both [3] and [18] derive competitiveness ratio guarantee while we derive sublinear regret upper bound; also we relax the non-negativeness assumptions on the entries $a_{ij}$ in the constraint matrix.

Recall that in Proposition 1, we establish the connection between the optimal solutions of the scaled LP and the original LP. Now we extend the result and connect history and future observations (under the random permutation model) in a more systematic way. The following proposition quantifies the difference of objective value or constraint consumption between the past and future observations based on the same dual vector $\boldsymbol{p}$, and the expected difference is roughly on the order of $\sqrt{\frac{m\log n}{\min\{t, n-t\}}}$.

**Proposition 2.** *If $\{(r_j, \boldsymbol{a}_j)\}_{j=1}^n$ is a random permutation of dataset $\mathcal{D}$ and satisfy Assumption 2, we have*

$$\mathbb{E}\left[\sup_{\boldsymbol{p}\geq 0}\left|\frac{1}{n-t}\sum_{j=t+1}^n r_j I(r_j > \boldsymbol{a}_j^\top \boldsymbol{p}) - \frac{1}{t}\sum_{j=1}^t r_j I(r_j > \boldsymbol{a}_j^\top \boldsymbol{p})\right|\right] \leq \frac{4\bar{r}}{\sqrt{\min\{t, n-t\}}} + \frac{2\sqrt{2\bar{r}^2 m \log n}}{\sqrt{\min\{t, n-t\}}}$$

$$\mathbb{E}\left[\sup_{\boldsymbol{p}\geq 0}\left|\frac{1}{n-t}\sum_{j=t+1}^n a_{ij} I(r_j > \boldsymbol{a}_j^\top \boldsymbol{p}) - \frac{1}{t}\sum_{j=1}^t a_{ij} I(r_j > \boldsymbol{a}_j^\top \boldsymbol{p})\right|\right] \leq \frac{4\bar{a}}{\sqrt{\min\{t, n-t\}}} + \frac{2\sqrt{2\bar{a}^2 m \log n}}{\sqrt{\min\{t, n-t\}}}$$

*for any $1 \leq i \leq m$, $1 \leq t \leq n-1$.*

The proof of the above proposition builds upon the notion of *permutational Rademacher Complexity* [27]. It first bounds the left hand with the permutational Rademacher Complexity of function class $\mathcal{F}_{\boldsymbol{p}} = \left\{f_{\boldsymbol{p}} : f_{\boldsymbol{p}}(r, \boldsymbol{a}) = rI\left(r > \boldsymbol{a}^T \boldsymbol{p}\right)\right\}$. Then it mimics the analysis in [27] and relates the permutational Rademacher Complexity with the conditional Rademacher Complexity of $\mathcal{F}$, while the latter has a natural upper bound based on Massart's Lemma.

## 5.1 Regret Bounds for Two "Slower" Algorithms

The Dynamic Learning Algorithm was first proposed in [3] and then refined in [20] (See Algorithm 2 in supplementary document). The idea is to construct a dual price $\boldsymbol{p}_t$ at each time $t$ based on solving a scaled LP problem ($s$-S-LP) with the first $t-1$ observations, and then to use $\boldsymbol{p}_t$ to decide the value of $x_t$. The algorithm is much slower than Algorithm 1 since at each iteration, an LP (of growing size) is solved to compute the dual price. For the analysis, the PRC theory presented earlier thus provides a machinery to relate the evaluation of $\boldsymbol{p}_t$ on the past $t-1$ samples with that of the incoming sample at time $t$.

**Theorem 3.** *Under Assumption 2 and 3, the regret and expected constraint violation of the Dynamic Learning Algorithm ([3, 20]) satisfy*

$$R_n^* - \mathbb{E}[R_n] \leq O\left(\sqrt{mn}\right)$$

$$\mathbb{E}[\boldsymbol{Ax} - \boldsymbol{b}] \leq O(\sqrt{mn}\log n)$$

*for all $m, n \in \mathbb{N}^+$ and $\mathcal{D} \in \Xi_D$. Here $\boldsymbol{x}$ is the output of the Dynamic Learning Algorithm.*

Theorem 3 shows that the regret and constraint violation can be reduced by a factor of $O(\sqrt{m})$ compared with Algorithm 1, with the price of computation cost.

The Primal-Beats-Dual Algorithm (See Algorithm 3 in the supplementary document) was proposed in [18] and it can be viewed as a primal version of the Dynamic Learning Algorithm. At each time $t$, it solves the primal scaled LP ($s$-S-LP) and projects the primal optimal solution $\tilde{x}_t^{(t)}$ to a binary value. Therefore it also involves solving an LP at each time period and is slower than Algorithm 1.

**Theorem 4.** *Under Assumption 2 and 3, the regret and expected constraint violation of the Primal-Beats-Dual Algorithm [18] satisfy*

$$R_n^* - \mathbb{E}[R_n] \leq O\left(\sqrt{mn}\right)$$

$$\mathbb{E}[v(\boldsymbol{x})] \leq O(\sqrt{mn}\log n)$$

*for all $m, n \in \mathbb{N}^+$ and $\mathcal{D} \in \Xi_D$.*

Theorem 4 states that the regret and constraint violation of Algorithm 3 are on the order of $O(\sqrt{mn})$. The analysis of objective value builds upon the Proposition 1 and the analysis of the constraint violation employs a backward super-Martingale argument. Like Algorithm 2, the extra computation cost here also helps improve the algorithm performance in terms of $m$.

The numerical experiments and the proofs for all the theorems in this paper are included in the supplementary materials. Also, we present a randomized approach to convert an infeasible solution to a feasible one with provable guarantee.

## Acknowledgement & Ethical Aspects

We thank the four anonymous reviewers whose comments/suggestions helped improve and clarify this manuscript, and thank all seminar participants at Stanford, NYU Stern, Columbia DRO, Chicago Booth, and Imperial College Business School for helpful discussions and comments.

Our paper takes a purely theoretical perspective, and the discussed model covers a wide range of applications including resource allocation, revenue management, assignment, and matching problems. As far as we can see, there is no potential ethical issue arising from the results developed in our paper.

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
