[Supplementary Material]

# Appendix

## A1 Two Algorithms in Literature

The two algorithm discussed in Section 5.1 are Algorithm 2 and Algorithm 3.

---

**Algorithm 2** Dynamic Learning Algorithm (first proposed in [3] and then refined in [20])

---

1: Input: $\boldsymbol{d}$
2: Let $\boldsymbol{p}_1 = \boldsymbol{0}$
3: **for** $t = 1, 2, ..., n$ **do**
4:
$$x_t = \begin{cases} 1, & \text{if } r_t > \boldsymbol{a}_t^\top \boldsymbol{p}_t \\ 0, & \text{if } r_t \le \boldsymbol{a}_t^\top \boldsymbol{p}_t \end{cases}$$

5:   The scaled primal LP is
$$\max \sum_{j=1}^{t} r_j x_j$$
$$\text{s.t. } \sum_{j=1}^{t} a_{ij} x_j \le t d_i, \quad i = 1, ..., m$$
$$0 \le x_j \le 1, \quad j = 1, ..., t$$

6:   Solve its dual problem and obtain the optimal dual variable $\boldsymbol{p}_{t+1}$
$$\boldsymbol{p}_{t+1} = \underset{\boldsymbol{p} \ge 0}{\arg\min} \ \sum_{i=1}^{m} d_i p_i + \frac{1}{t} \sum_{j=1}^{t} \left( r_j - \sum_{i=1}^{m} a_{ij} p_i \right)^+$$

7: **end for**

---

---

**Algorithm 3** Primal-beats-dual Algorithm [18]

---

1: Input: $\boldsymbol{d}$
2: Let $\boldsymbol{p}_1 = \boldsymbol{0}$
3: **for** $t = 1, 2, ..., n$ **do**
4:   Solve the scaled LP
$$\max \sum_{j=1}^{t} r_j x_j$$
$$\text{s.t. } \sum_{j=1}^{t} a_{ij} x_j \le t d_i, \quad i = 1, ..., m$$
$$0 \le x_j \le 1, \quad j = 1, ..., t$$

5:   Denote the optimal solution as $\tilde{\boldsymbol{x}}^{(t)} = (\tilde{x}_1^{(t)}, ..., \tilde{x}_t^{(t)})$
6:
$$x_t = \begin{cases} 1, & \text{with probability } \tilde{x}_t^{(t)} \\ 0, & \text{with probability } 1 - \tilde{x}_t^{(t)} \end{cases}$$

7: **end for**

---

## B1   Concentration Inequalities under Random Permutation

**Lemma 4.** *Let $U_1, ..., U_n$ be a random sample without replacement from the real numbers $\{c_1, ..., c_N\}$. Then for every $s > 0$,*

$$
\mathbb{P}(|\bar{U}_n - \bar{c}_N| \geq s) \leq
\begin{cases}
2\exp\left(-\frac{2ns^2}{\Delta_N^2}\right) & \text{(Hoeffding)}, \\
2\exp\left(-\frac{2ns^2}{(1-(n-1)/N)\Delta_N^2}\right) & \text{(Serfling)}, \\
2\exp\left(-\frac{ns^2}{2\sigma_N^2 + s\Delta_N}\right) & \text{(Hoeffding-Bernstein)}, \\
2\exp\left(-\frac{ns^2}{m\sigma_N^2}\right) & \text{if } N = mn \text{ (Massart)},
\end{cases}
$$

*where $\bar{c}_N = \frac{1}{N}\sum\limits_{i=1}^{N} c_i$, $\sigma_N^2 = \frac{1}{N}\sum\limits_{i=1}^{N}(c_i - \bar{c}_N)^2$ and $\Delta_N = \max\limits_{1 \leq i \leq N} c_i - \min\limits_{1 \leq i \leq N} c_i$.*

*Proof.* See Theorem 2.14.19 in [28]. $\qquad\square$

## B2   Proof of Lemma 1

*Proof.* By taking expectation with respect to the elements in (5), the function $f_n(\boldsymbol{p})$ in (5) can be viewed as a *sample average approximation* of the following stochastic program

$$
\min_{\boldsymbol{p}} f(\boldsymbol{p}) = \boldsymbol{d}^\top \boldsymbol{p} + \mathbb{E}_{(r,\boldsymbol{a}) \sim \mathcal{P}}\left[\left(r - \boldsymbol{a}^\top \boldsymbol{p}\right)^+\right] \tag{SP}
$$
$$
\text{s.t. } \boldsymbol{p} \geq \boldsymbol{0}.
$$

Denote the optimal solution to (SP) as $\boldsymbol{p}^*$. For $\boldsymbol{p}^*$, we have

$$
\underline{d}\|\boldsymbol{p}^*\|_1 \leq \boldsymbol{d}^T \boldsymbol{p}^* \overset{(a)}{\leq} \mathbb{E}r \leq \bar{r},
$$

where inequality (a) is due to that if otherwise, $\boldsymbol{p}^*$ cannot be the optimal solution because it will give a larger objective value of $f(\boldsymbol{p})$ than setting $\boldsymbol{p} = \boldsymbol{0}$. Given the non-negativeness of $\boldsymbol{p}^*$, we have $\|\boldsymbol{p}^*\|_2 \leq \|\boldsymbol{p}^*\|_1$. So we obtain the first inequality in the lemma.

For $\boldsymbol{p}_t$ specified by Algorithm 1, we have,

$$
\begin{aligned}
\|\boldsymbol{p}_{t+1}\|_2^2 &\leq \|\boldsymbol{p}_t + \gamma_t\left(\boldsymbol{a}_t x_t - \boldsymbol{d}\right)\|_2^2 \\
&= \|\boldsymbol{p}_t\|_2^2 + \gamma_t^2\|\boldsymbol{a}_t x_t - \boldsymbol{d}\|_2^2 + 2\gamma_t(\boldsymbol{a}_t x_t - \boldsymbol{d})^\top \boldsymbol{p}_t \\
&\leq \|\boldsymbol{p}_t\|_2^2 + \gamma_t^2 m(\bar{a} + \bar{d})^2 + 2\gamma_t \boldsymbol{a}_t^\top \boldsymbol{p}_t x_t - 2\gamma_t \boldsymbol{d}^\top \boldsymbol{p}_t
\end{aligned}
$$

where the first inequality comes from the projection (into the non-negative orthant) step in the algorithm. Note that

$$
\boldsymbol{a}_t^\top \boldsymbol{p}_t x_t = \boldsymbol{a}_t^\top \boldsymbol{p}_t I(r_t > \boldsymbol{a}_t^\top \boldsymbol{p}_t) \leq r_t \leq \bar{r}.
$$

Therefore,

$$
\|\boldsymbol{p}_{t+1}\|_2^2 \leq \|\boldsymbol{p}_t\|_2^2 + \gamma_t^2 m(\bar{a} + \bar{d})^2 + 2\gamma_t \bar{r} - 2\gamma_t \boldsymbol{d}^\top \boldsymbol{p}_t,
$$

and it holds with probability 1.

Next, we establish that when $\|\boldsymbol{p}_t\|_2$ is large enough, then it must hold that $\|\boldsymbol{p}_{t+1}\|_2 \leq \|\boldsymbol{p}_t\|_2$. Specifically, when $\|\boldsymbol{p}_t\|_2 \geq \frac{2\bar{r} + m(\bar{a}+\bar{d})^2}{\underline{d}}$, we have

$$
\begin{aligned}
\|\boldsymbol{p}_{t+1}\|_2^2 - \|\boldsymbol{p}_t\|_2^2 &\leq \gamma_t^2 m(\bar{a} + \bar{d})^2 + 2\gamma_t \bar{r} - 2\gamma_t \boldsymbol{d}^\top \boldsymbol{p}_t \\
&\leq \gamma_t^2 m(\bar{a} + \bar{d})^2 + 2\gamma_t \bar{r} - 2\gamma_t \underline{d}\|\boldsymbol{p}_t\|_1 \\
&\leq \gamma_t^2 m(\bar{a} + \bar{d})^2 + 2\gamma_t \bar{r} - 2\gamma_t \underline{d}\|\boldsymbol{p}_t\|_2 \\
&\leq 0
\end{aligned}
$$

when $\gamma_t \leq 1$. On the other hand, when $\|\boldsymbol{p}_t\|_2 \leq \frac{2\bar{r} + m(\bar{a}+\bar{d})^2}{\underline{d}}$,

$$
\begin{aligned}
\|\boldsymbol{p}_{t+1}\|_2 &\leq \|\boldsymbol{p}_t + \gamma_t\left(\boldsymbol{a}_t x_t - \boldsymbol{d}\right)\|_2 \\
&\overset{(b)}{\leq} \|\boldsymbol{p}_t\|_2 + \gamma_t\|\boldsymbol{a}_t x_t - \boldsymbol{d}\|_2 \\
&\leq \frac{2\bar{r} + m(\bar{a} + \bar{d})^2}{\underline{d}} + m(\bar{a} + \bar{d})
\end{aligned}
$$

where (b) comes from the triangle inequality of the norm.

Combining these two scenarios with the fact that $\boldsymbol{p}_1 = \boldsymbol{0}$, we have

$$\|\boldsymbol{p}_t\|_2 \le \frac{2\bar{r} + m(\bar{a} + \bar{d})^2}{\underline{d}} + m(\bar{a} + \bar{d})$$

for $t = 1, ..., n$ with probability 1.

$\square$

## B3  Proof of Theorem 1

*Proof.* First, the primal optimal objective value is no greater than the dual objective with $\boldsymbol{p} = \boldsymbol{p}^*$. Specifically,

$$R_n^* = \text{P-LP} = \text{D-LP}$$
$$\le n\boldsymbol{d}^\top \boldsymbol{p}^* + \sum_{j=1}^{n} \left(r_j - \boldsymbol{a}_j^\top \boldsymbol{p}^*\right)^+.$$

Taking expectation on both sides,

$$\mathbb{E}\left[R_n^*\right] \le \mathbb{E}\left[n\boldsymbol{d}^\top \boldsymbol{p}^* + \sum_{t=1}^{n} \left(r_t - \boldsymbol{a}_t^\top \boldsymbol{p}^*\right)^+\right]$$
$$\le nf(\boldsymbol{p}^*).$$

Thus, the optimal objective value of the stochastic program (by a factor of $n$) is an upper bound for the expected value of the primal optimal objective. Hence

$$\mathbb{E}[R_n^* - R_n] \le nf(\boldsymbol{p}^*) - \sum_{j=1}^{n} \mathbb{E}\left[r_t I(r_t > \boldsymbol{a}_t^\top \boldsymbol{p}_t)\right]$$
$$\le \sum_{t=1}^{n} \mathbb{E}\left[f(\boldsymbol{p}_t)\right] - \sum_{t=1}^{n} \mathbb{E}\left[r_t I(r_t > \boldsymbol{a}_t^\top \boldsymbol{p}_t)\right]$$
$$\le \sum_{t=1}^{n} \mathbb{E}\left[\boldsymbol{d}^\top \boldsymbol{p}_t + \left(r_t - \boldsymbol{a}_t^\top \boldsymbol{p}_t\right)^+ - r_t I(r_t > \boldsymbol{a}_t^\top \boldsymbol{p}_t)\right]$$
$$= \sum_{t=1}^{n} \mathbb{E}\left[\left(\boldsymbol{d}^\top - \boldsymbol{a}_t I(r_t > \boldsymbol{a}_t^\top \boldsymbol{p}_t)\right)^\top \boldsymbol{p}_t\right].$$

where the expectation is taken with respect to $(r_t, \boldsymbol{a}_t)$'s. In above, the second line comes from the optimality of $\boldsymbol{p}^*$, while the third line is valid because of the independence between $\boldsymbol{p}_t$ and $(r_t, \boldsymbol{a}_t)$.

The importance of the above inequality lies in that it relates and represents the primal optimality gap in the dual prices $\boldsymbol{p}_t$ – which is the core of Algorithm 1. From the updating formula in Algorithm 1, we know

$$\|\boldsymbol{p}_{t+1}\|_2^2 \le \|\boldsymbol{p}_t\|_2^2 - \frac{2}{\sqrt{n}}\left(\boldsymbol{d} - \boldsymbol{a}_t I(r_t > \boldsymbol{a}_t^\top \boldsymbol{p}_t)\right)^\top \boldsymbol{p}_t + \frac{1}{n}\left\|\boldsymbol{d} - \boldsymbol{a}_t I(r_t > \boldsymbol{a}_t^\top \boldsymbol{p}_t)\right\|_2^2$$
$$\le \|\boldsymbol{p}_t\|_2^2 - \frac{2}{\sqrt{n}}\left(\boldsymbol{d} - \boldsymbol{a}_t I(r_t > \boldsymbol{a}_t^\top \boldsymbol{p}_n)\right)^\top \boldsymbol{p}_t + \frac{m(\bar{a} + \bar{d})^2}{n}.$$

Moving the cross-term to the right-hand-side, we have

$$\sum_{t=1}^{n} \left(\boldsymbol{d} - \boldsymbol{a}_t I(r_t > \boldsymbol{a}_t^\top \boldsymbol{p}_t)\right)^\top \boldsymbol{p}_t \le \sum_{t=1}^{n} \left(\sqrt{n}\|\boldsymbol{p}_t\|_2^2 - \sqrt{n}\|\boldsymbol{p}_{t+1}\|_2^2 + \frac{m(\bar{a} + \bar{d})^2}{\sqrt{n}}\right)$$
$$\le m(\bar{a} + \bar{d})^2 \sqrt{n}.$$

Consequently,

$$\mathbb{E}[R_n^* - R_n] \le m(\bar{a} + \bar{d})^2 \sqrt{n}$$

hold for all $n$ and distribution $\mathcal{P} \in \Xi$.

For the constraint violation, if we revisit the updating formula, we have

$$\boldsymbol{p}_{t+1} \ge \boldsymbol{p}_t + \frac{1}{\sqrt{n}}\left(\boldsymbol{a}_t x_t - \boldsymbol{d}\right)$$

where the inequality is elementwise. Therefore,

$$\sum_{t=1}^{n} \boldsymbol{a}_t x_t \leq n\boldsymbol{d} + \sum_{t=1}^{n} \sqrt{n}(\boldsymbol{p}_{t+1} - \boldsymbol{p}_t)$$
$$\leq \boldsymbol{b} + \sqrt{n}\boldsymbol{p}_{n+1}$$

In the second line, we remove the term involve $\boldsymbol{p}_1$ with the algorithm specifying $\boldsymbol{p}_1 = \boldsymbol{0}$. Then with Lemma 1, we have

$$\mathbb{E}\left[v(\boldsymbol{x})\right] = \mathbb{E}\left[\|\left(\boldsymbol{A}\boldsymbol{x} - \boldsymbol{b}\right)^+\|_2\right] \leq \sqrt{n}\mathbb{E}\|\boldsymbol{p}_{n+1}\|_2 \leq \left(\frac{2\bar{r} + m(\bar{a} + \bar{d})^2}{\underline{d}} + m(\bar{a} + \bar{d})\right)\sqrt{n}.$$

$\square$

## B4    Proof of Proposition 1

*Proof.* Define $\mathrm{SLP}(s, \boldsymbol{b}_0)$ as the following LP

$$\max \quad \sum_{j=1}^{s} r_j x_j$$
$$\text{s.t.} \quad \sum_{j=1}^{s} a_{ij} x_j \leq \frac{sb_i}{n} + b_{0i}$$
$$0 \leq x_j \leq 1 \text{ for } j = 1, ..., s.$$

where $\boldsymbol{b}_0 = (b_{01}, ..., b_{0m})$ denotes the constraint relaxation quantity for the scaled LP. Denote the optimal objective value of $\mathrm{SLP}(s, \boldsymbol{b}_0)$ as $R^*(s, \boldsymbol{b}_0)$. Also, denote $\boldsymbol{x}(\boldsymbol{p}) = (x_1(\boldsymbol{p}), ..., x_n(\boldsymbol{p}))$ and $x_j(\boldsymbol{p}) = I(r_j > \boldsymbol{a}_j^\top \boldsymbol{p})$. It denotes the decision variables we obtain with a dual price $\boldsymbol{p}$.

We prove the following three results:

(i)  The following bounds hold for $R_n^*$,

$$\sum_{j=1}^{n} r_j x_j(\boldsymbol{p}_n^*) \leq R_n^* \leq \sum_{j=1}^{n} r_j x_j(\boldsymbol{p}_n^*) + m\bar{r}.$$

(ii)  When $s \geq \max\{16\bar{a}^2, \exp\{16\bar{a}^2\}, e\}$, then the optimal dual solution $\boldsymbol{p}_n^*$ is a feasible solution to $\mathrm{SLP}\left(s, \frac{\log s}{\sqrt{s}}\boldsymbol{1}\right)$ with probability no less than $1 - \frac{m}{s}$.

(iii)  The following inequality holds for the optimal objective values to the scaled LP and its relaxation

$$R_s^* \geq R^*\left(s, \frac{\log s}{\sqrt{s}}\boldsymbol{1}\right) - \frac{\bar{r}\sqrt{s}\log s}{\underline{d}}.$$

**For part (i)**, this inequality replace the optimal value with bounds containing the objective values obtained by adopting optimal dual solution. The left hand side of the inequality comes from the complementarity condition while the right hand side can be shown from Lemma 2.

**For part (ii)**, the motivation to introduce a relaxed form of the scaled LP is to ensure the feasibility of $\boldsymbol{p}_n^*$. The key idea for the proof is to utilize the feasibility of $\boldsymbol{p}_n^*$ for (2). To see that, let $\alpha_{ij} = a_{ij} I(r_j > \boldsymbol{a}_j^T \boldsymbol{p}^*)$ and

$$c_\alpha = \max_{i,j} \alpha_{ij} - \min_{i,j} \alpha_{ij} \leq 2\bar{a},$$
$$\bar{\alpha}_i = \frac{1}{n} \sum_{j=1}^{n} \alpha_{ij} = \frac{1}{n} \sum_{j=1}^{n} a_{ij} x_t(\boldsymbol{p}_n^*) \leq d_i, \qquad (7)$$
$$\sigma_i^2 = \frac{1}{n} \sum_{j=1}^{n} (\alpha_{ij} - \bar{\alpha}_i)^2 \leq 4\bar{a}^2.$$

Here the first and third inequality comes from the bounds on $a_{ij}$'s while the second one comes from the feasibility of the optimal solution for (2).

Then, when $k > \max\{16\bar{a}^2, \exp\{16\bar{a}^2\}, e\}$, by applying Hoeffding-Bernstein's Inequality

$$\mathbb{P}\left(\sum_{j=1}^{k}\alpha_{ij} - kd_i \geq \sqrt{k}\log k\right) \overset{(e)}{\leq} \mathbb{P}\left(\sum_{j=1}^{k}\alpha_{ij} - k\bar{\alpha}_i \geq \sqrt{k}\log k\right)$$

$$\overset{(f)}{\leq} \exp\left(-\frac{k\log^2 k}{8k\bar{a}^2 + 2\bar{a}\sqrt{k}\log k}\right)$$

$$\overset{(g)}{\leq} \frac{1}{k}$$

for $i = 1, ..., m$. Here inequality (e) comes from (7), (f) comes from applying Lemma 4, and (g) holds when $s > \max\{16\bar{a}^2, \exp\{16\bar{a}^2\}, e\}$.

Let event

$$E_i = \left\{\sum_{j=1}^{s}\alpha_{ij} - sd_i < \sqrt{s}\log s\right\}$$

and $E = \bigcap_{i=1}^{m} E_i$. The above derivation tells $\mathbb{P}(E_i) \geq 1 - \frac{1}{s}$ By applying union bound, we obtain $\mathbb{P}(E) \geq 1 - \frac{m}{s}$ and it completes the proof of part (ii).

**For part(iii)**, denote the optimal solution to $\text{SLP}\left(s, \frac{\log s}{\sqrt{s}}\mathbf{1}\right)$ as $\tilde{\boldsymbol{p}}_s$.

$$R^*\left(s, \frac{\log s}{\sqrt{s}}\mathbf{1}\right) = s\left(\boldsymbol{d} + \frac{\log s}{\sqrt{s}}\mathbf{1}\right)^\top\tilde{\boldsymbol{p}}_s^* + \sum_{j=1}^{s}\left(r_j - \boldsymbol{a}_j^\top\tilde{\boldsymbol{p}}_s^*\right)^+$$

$$\leq s\left(\boldsymbol{d} + \frac{\log s}{\sqrt{s}}\mathbf{1}\right)^\top\boldsymbol{p}_s^* + \sum_{j=1}^{s}\left(r_j - \boldsymbol{a}_j^\top\boldsymbol{p}_s^*\right)^+$$

$$\leq \frac{\bar{r}\sqrt{s}\log s}{\underline{d}} + R_s^*.$$

where the first inequality comes from dual optimality of $\tilde{\boldsymbol{p}}_s^*$ and the second inequality comes from the upper bound of $\|\boldsymbol{p}_s^*\|$ and the duality of the scaled LP $R_s^*$. Therefore,

$$R_s^* \geq R^*\left(s, \frac{\log s}{\sqrt{s}}\mathbf{1}\right) - \frac{\bar{r}\sqrt{s}\log s}{\underline{d}}.$$

Finally, we complete the proof with the help of the above three results.

$$\frac{1}{s}\mathbb{E}\left[\mathbb{I}_E R_s^*\right] \geq \frac{1}{s}\mathbb{E}\left[\mathbb{I}_E R^*\left(s, \frac{\log s}{\sqrt{s}}\mathbf{1}\right)\right] - \frac{\bar{r}\sqrt{s}\log s}{\underline{d}}$$

$$\geq \frac{1}{s}\mathbb{E}\left[\mathbb{I}_E\sum_{j=1}^{s}r_j x_j(\boldsymbol{p}^*)\right] - \frac{\bar{r}\sqrt{s}\log s}{\underline{d}}$$

where $\mathbb{I}_E$ denotes an indicator function for event $E$. The first line comes from applying part (iii) while the second line comes from the feasibility of $\boldsymbol{p}^*$ on event $E$. Then,

$$\frac{1}{s}\mathbb{E}\left[R_s^*\right] \geq \frac{1}{s}\mathbb{E}\left[\sum_{j=1}^{s}r_j x_j(\boldsymbol{p}^*)\right] - \frac{\bar{r}\sqrt{s}\log s}{\underline{d}} - \frac{m\bar{r}}{s}$$

$$= \frac{1}{n}\mathbb{E}\left[\sum_{j=1}^{n}r_j x_j(\boldsymbol{p}^*)\right] - \frac{\bar{r}\sqrt{s}\log s}{\underline{d}} - \frac{m\bar{r}}{s}$$

$$\geq \frac{1}{n}R_n^* - \frac{\bar{r}\sqrt{s}\log s}{\underline{d}} - \frac{m\bar{r}}{s} - \frac{m\bar{r}}{n}$$

where the first line comes from part (ii) – the probability bound on event $E$, the second line comes from the symmetry of the random permutation probability space, and the third line comes from part (i). We complete the proof. $\qquad\square$

## B5 Proof of Theorem 2

*Proof.* For the regret bound,

$$R_n^* - \mathbb{E}\left[R_n\right] = R_n^* - \sum_{t=1}^{n}\mathbb{E}\left[r_t x_t\right]$$

where $x_t$'s are specified according to Algorithm 1. Then

$$R_n^* - \mathbb{E}[R_n] = R_n^* - \sum_{t=1}^n \frac{1}{t}\mathbb{E}[R_t^*] + \sum_{t=1}^n \frac{1}{t}\mathbb{E}[R_t^*] - \sum_{t=1}^n \mathbb{E}[r_t x_t]$$

$$= \sum_{t=1}^n \left(\frac{1}{n}R_n^* - \frac{1}{t}\mathbb{E}[R_t^*]\right) + \sum_{t=1}^n \mathbb{E}\left[\frac{1}{n+1-t}\tilde{R}_{n-t+1}^* - r_t x_t\right] \quad (8)$$

where $\tilde{R}_{n-t+1}^*$ is defined as the optimal value of the following LP

$$\max \; \sum_{j=t}^n r_j x_j$$

$$\text{s.t.} \; \sum_{j=t}^n a_{ij} x_j \leq \frac{(n-t+1)b_i}{n}$$

$$0 \leq x_j \leq 1 \; \text{for } j = 1, ..., m.$$

For the first part of (8), we can apply Proposition 1. Meanwhile, the analyses of the second part takes a similar form as the previous stochastic input model. Specifically,

$$\mathbb{E}\left[\frac{1}{n+1-t}\tilde{R}_{n-t+1}^* - r_t x_t\right] \leq \left(\boldsymbol{d} - \boldsymbol{a}_t I(r_t > \boldsymbol{a}_t^\top \boldsymbol{p}_t)\right)^\top \boldsymbol{p}_t.$$

Similar to the stochastic input model,

$$\|\boldsymbol{p}_{t+1}\|_2^2 \leq \|\boldsymbol{p}_t\|_2^2 - \frac{2}{\sqrt{n}}\left(\boldsymbol{d} - \boldsymbol{a}_t I(r_t > \boldsymbol{a}_t^\top \boldsymbol{p}_t)\right)^\top \boldsymbol{p}_t + \frac{1}{n}\left\|\boldsymbol{d} - \boldsymbol{a}_t I(r_t > \boldsymbol{a}_t^\top \boldsymbol{p}_t)\right\|_2^2$$

$$\leq \|\boldsymbol{p}_t\|_2^2 - \frac{2}{\sqrt{n}}\left(\boldsymbol{d} - \boldsymbol{a}_t I(r_t > \boldsymbol{a}_t^\top \boldsymbol{p}_t)\right)^\top \boldsymbol{p}_t + \frac{m(\bar{a}+\bar{d})^2}{n}.$$

Thus, we have

$$\sum_{t=1}^n \mathbb{E}\left[\left(\boldsymbol{d} - \boldsymbol{a}_t I(r_t > \boldsymbol{a}_t^\top \boldsymbol{p}_t)\right)^\top \boldsymbol{p}_t\right] \leq \sum_{t=1}^n \mathbb{E}\left[\sqrt{n}(\|\boldsymbol{p}_t\|_2^2 - \|\boldsymbol{p}_{t+1}\|_2^2)\right] + \sum_{t=1}^n \frac{m(\bar{a}+\bar{d})^2}{\sqrt{n}}$$

$$\leq m(\bar{a}+\bar{d})^2\sqrt{n}.$$

Combine two parts above, finally we have

$$R_n^* - \mathbb{E}[R_n(\pi)] \leq m\bar{r} + \frac{\bar{r}\log n\sqrt{n}}{\underline{d}} + m\bar{r}\log n + \frac{\max\{16\bar{a}^2, \exp\{16\bar{a}^2\}, e\}\bar{r}}{n} + m(\bar{a}+\bar{d})^2\sqrt{n}$$

$$= O((m+\log n)\sqrt{n})$$

Thus, we complete the proof for the regret. The proof for the constraint violation part follows exactly the same way as the stochastic input model. □

## B6 Proof of Proposition 2

First, we define the notion of permutational Rademacher complexity. Consider set $\mathcal{Z}_n = \{z_1, ..., z_n\}$ where $z_j \in \mathbb{R}^k, j = 1, ..., n$ and a family of functions $\mathcal{F} = \{f : \mathbb{R}^k \to \mathbb{R}\}$ (to be specified later). Throughout this section, we use the subscript to indicate the cardinality of a set. For function $f \in \mathcal{F}$ and $\mathcal{S} \subset \mathcal{Z}$, denote $\bar{f}(\mathcal{S}) = \frac{1}{|\mathcal{S}|}\sum_{x\in\mathcal{S}} f(x)$ as the mean function value on the set $\mathcal{S}$. The definition of the permutational Rademacher complexity and its analysis largely mimic the analyses of the transductive learning problem in [27].

**Definition 1** (Permutational Rademacher Complexity and Conditional Rademacher Complexity (See Definition 3 in [27])). *For any $1 \leq s \leq t - 1$, permutational Rademacher complexity (PRC) is defined as follows:*

$$Q_{t,s}(\mathcal{F}, \mathcal{Z}_t) = \mathbb{E}\sup_{f\in\mathcal{F}}\left|\bar{f}(\mathcal{Z}_s) - \bar{f}(\tilde{\mathcal{Z}}_l)\right|,$$

*where $\mathcal{Z}_s$ is subset of $\mathcal{Z}_t$ with $s$ elements sampled uniformly without replacement and $\tilde{\mathcal{Z}}_l = \mathcal{Z}_t \backslash \mathcal{Z}_s$, $l = t - s$. The expectation is taken with respect to the random sampling of $\mathcal{Z}_s$.*

*Conditional Rademacher complexity (CRC) is defined as follows:*

$$R_t(\mathcal{F}, \mathcal{Z}_t) = \mathbb{E}\sup_{f\in\mathcal{F}}\left|\frac{1}{t}\sum_{j=1}^t \epsilon_j f(z_j)\right|,$$

*where $\mathcal{Z}_t = \{z_1, ..., z_t\}$ and $\epsilon_j$'s are i.i.d. random variables following Rademacher distribution ($P(\epsilon_j = 1) = P(\epsilon_j = -1) = 1/2$). The expectation here is taken with respect to $\epsilon_j$'s.*

Both the above two quantities are dependent on the set $\mathcal{Z}_t$ because for the PRC, the two subsets $\mathcal{Z}_s$ and $\tilde{\mathcal{Z}}_l$ are sampled from $\mathcal{Z}_t$ and for CRC, it is computed based on the function values of the elements in $\mathcal{Z}_t$. Both PRC and CRC are deterministic with a given function class $\mathcal{F}$ and conditional on $\mathcal{Z}_t$. However, they could be random variables if the set $\mathcal{Z}_t$ is random. The following lemma explains the motivation for the definition of permutational Rademacher complexity and it is inspired from Theorem 2 in [27].

**Lemma 5.** *$\mathcal{Z}_t$ is a subset of $\mathcal{Z}_n$ obtained by uniform sampling without replacement, and $\tilde{\mathcal{Z}}_{t'} = \mathcal{Z}_n \backslash \mathcal{Z}_t$, $t' = n - t$. Without loss of generality, assume $t \geq t'$, then the following inequality holds for all $s < t'$,*

$$\mathbb{E}\left[\sup_{f \in \mathcal{F}} \left|\bar{f}(\mathcal{Z}_t) - \bar{f}(\tilde{\mathcal{Z}}_{t'})\right| \Big| \mathcal{Z}_n\right] \leq \mathbb{E}\left[Q_{t,s}(\mathcal{F}, \mathcal{Z}_t) \Big| \mathcal{Z}_n\right]$$

*where the expectation is taken with respect to the random sampling of $\mathcal{Z}_t$ from $\mathcal{Z}_n$*

*Proof.* We have

$$\mathbb{E}\left[\sup_{f \in \mathcal{F}} \left|\bar{f}(\mathcal{Z}_t) - \bar{f}(\tilde{\mathcal{Z}}_{t'})\right| \Big| \mathcal{Z}_n\right] = \mathbb{E}\left[\sup_{f \in \mathcal{F}} \left|\mathbb{E}\left[\bar{f}(\mathcal{Z}_{t-s})|\mathcal{Z}_t\right] - \mathbb{E}\left[\bar{f}(\tilde{\mathcal{Z}}_s)|\tilde{\mathcal{Z}}_{t'}\right]\right| \Big| \mathcal{Z}_n\right]$$

$$\leq \mathbb{E}\left[\sup_{f \in \mathcal{F}} \left|\bar{f}(\mathcal{Z}_{t-s}) - \bar{f}(\tilde{\mathcal{Z}}_s)\right| \Big| \mathcal{Z}_n\right] = \mathbb{E}\left[Q_{t,s}(\mathcal{F}, \mathcal{Z}_t) \big| \mathcal{Z}_n\right].$$

For the first line, on the right hand side, the two inner expectations are taken with respect to a uniform random sampling on $\mathcal{Z}_t$ and $\tilde{\mathcal{Z}}_{t'}$ respectively. Specifically, $\mathcal{Z}_{t-s}$ (or $\tilde{\mathcal{Z}}_s$) can be viewed as a random sampled subset from $\mathcal{Z}_t$ (or $\tilde{\mathcal{Z}}_{t'}$). For the second line, the first part comes from Jensen's inequality and the expectation in the second part is taken with respect to the random sampling of $\mathcal{Z}_t$ from $\mathcal{Z}_n$. $\square$

Let $\mathcal{F}_{\boldsymbol{p}} = \left\{f_{\boldsymbol{p}} : f_{\boldsymbol{p}}(r, \boldsymbol{a}) = rI\left(r > \boldsymbol{a}^T \boldsymbol{p}\right)\right\}$ denote a family of functions $f : \mathbb{R}^{m+1} \rightarrow \mathbb{R}$ indexed by the parameter $\boldsymbol{p}$, and let $(r_1, \boldsymbol{a}_1), ...., (r_n, \boldsymbol{a}_n)$ be a random permutation of the dataset $\mathcal{D}$. Denote $\mathcal{Z}_t = \{(r_1, \boldsymbol{a}_1), ..., (r_t, \boldsymbol{a}_t)\}$, and then $\mathcal{Z}_t$ can be viewed as a randomly sampled subset of $\mathcal{D}$. The following lemma relates the PRC with the classic notion of CRC, and the benefit is that the CRC under random permutation model possesses a natural upper bound.

**Lemma 6.** *The following inequalities hold for PRC and CRC of the family $\mathcal{F}_{\boldsymbol{p}}$ and dataset $\mathcal{D}$ that satisfies Assumption 2,*

$$\left|Q_{t,\lfloor t/2 \rfloor}(\mathcal{F}_{\boldsymbol{p}}, \mathcal{Z}_t) - R_t(\mathcal{F}_{\boldsymbol{p}}, \mathcal{Z}_t)\right| \leq \frac{4\bar{r}}{\sqrt{t}}.$$

$$R_t(\mathcal{F}, Z_t) \leq \frac{\sqrt{2\bar{r}^2 m \log n}}{\sqrt{t}}$$

*Proof.* For the first inequality, we refer to Theorem 3 in [27]. For the second inequality, it is a direct application of Massart's Lemma (See Lemma 26.8 of [26]). $\square$

*Proof of Proposition 2.* Let $\mathcal{F}_{\boldsymbol{p}} = \left\{f_{\boldsymbol{p}} : f_{\boldsymbol{p}}(r, \boldsymbol{a}) = rI\left(r > \boldsymbol{a}^T \boldsymbol{p}\right)\right\}$, $\mathcal{Z}_t = \{(r_1, \boldsymbol{a}_1), ..., (r_t, \boldsymbol{a}_t)\}$, and $\tilde{Z}_{n-t} = \{(r_{t+1}, \boldsymbol{a}_{t+1}), ..., (r_n, \boldsymbol{a}_n)\}$. Also, we assume $n - t > t$ without loss of generality. Then,

$$\mathbb{E}\left[\left|\frac{1}{n-t}\sum_{j=t+1}^{n} r_j I(r_j > \boldsymbol{a}_j^\top \boldsymbol{p}) - \frac{1}{t}\sum_{j=1}^{t} r_j I(r_j > \boldsymbol{a}_j^\top \boldsymbol{p})\right|\right] \leq \mathbb{E}\left[\sup_{f \in \mathcal{F}_{\boldsymbol{p}}} \left|\bar{f}(\mathcal{Z}_t) - \bar{f}(\tilde{\mathcal{Z}}_{n-t})\right| \Big| \mathcal{Z}_n = \mathcal{D}\right]$$

$$\leq \mathbb{E}\left[Q_{t,\lfloor t/2 \rfloor}(\mathcal{F}, \mathcal{Z}_t) \Big| \mathcal{Z}_n\right]$$

$$\leq \frac{4\bar{r}}{t} + \frac{2\sqrt{2\bar{r}^2 m \log n}}{\sqrt{t}}.$$

Here the first line comes from taking maximum over $\mathcal{F}_{\boldsymbol{p}}$, the second line comes from lemma 5 and the third line comes from lemma 6.

Similarly, we can show that the inequality on $a_{ij}$'s holds. Thus the proof is completed. $\square$

## B6.1 Proof for Theorem 3

*Proof.* At time $t + 1$,

$$\mathbb{E}\left[r_{t+1} x_{t+1}\right] = \mathbb{E}\left[r_{t+1} I(r_{t+1} > \boldsymbol{a}_{t+1}^\top \boldsymbol{p}_{t+1})\right]$$

$$= \frac{1}{n-t}\mathbb{E}\left[\sum_{j=t+1}^{n} r_j I(r_j > \boldsymbol{a}_j \boldsymbol{p}_{t+1})\right]$$

$$\geq \frac{1}{t}\mathbb{E}\left[\sum_{j=1}^{t} r_j I(r_j > \boldsymbol{a}_j \boldsymbol{p}_{t+1})\right] - \frac{4\bar{r}}{\sqrt{\min\{t, n-t\}}} - \frac{2\sqrt{2\bar{r}^2 m \log n}}{\sqrt{\min\{t, n-t\}}},$$

where the expectation is taken with respect to the random permutation. The first line comes from the algorithm design, the second line comes from the symmetry over the last $n - t$ terms, and the last line comes from the application of Proposition 2. To relate the first term in the last line with the offline optimal $R_n^*$, we utilize Proposition 1. Then the optimality gap of Algorithm 2 is as follows,

$$R_n^* - \mathbb{E}\left[\sum_{t=1}^{n} r_t x_t\right] = R_n^* - \sum_{t=1}^{n} \mathbb{E}\left[r_t x_t\right]$$

$$\leq R_n^* - \sum_{t=2}^{n}\left(\frac{1}{t}\mathbb{E}\left[\sum_{j=1}^{t} r_j I(r_j > \boldsymbol{a}_j \boldsymbol{p}_t)\right] - \frac{4\bar{r} + 2\sqrt{2\bar{r}^2 m \log n}}{\sqrt{\min\{t, n-t\}}}\right)$$

$$\leq m\bar{r} + \frac{\bar{r}}{\underline{d}}\sqrt{n}\log n + m\bar{r}\log n + \bar{r}\max\{16\bar{a}^2, e^{16\bar{a}^2}, e\} + \left(8\bar{r} + 4\sqrt{2\bar{r}^2 m \log n}\right)\sqrt{n}$$

$$= O(\sqrt{mn}\log n)$$

where the last line comes from an application of Proposition 1. Next, we analyze the constraint; again, from Proposition 2, we know

$$\frac{1}{n-t}\mathbb{E}\left[\sum_{j=t+1}^{n} a_{ij} I(r_j > \boldsymbol{a}_j^\top \boldsymbol{p}_t)\right] \leq \mathbb{E}\left[\frac{1}{t}\sum_{j=1}^{t} a_{ij} I(r_j > \boldsymbol{a}_j^\top \boldsymbol{p}_t)\right] + \frac{4\bar{a}}{\sqrt{\min\{t, n-t\}}} + \frac{2\sqrt{2\bar{a}^2 m \log n}}{\sqrt{\min\{t, n-t\}}}$$

$$\leq d_i + \frac{6\sqrt{2\bar{a}^2 m \log n}}{\sqrt{\min\{t, n-t\}}}$$

where the second line comes from the feasibility of the scaled LP solved at time $t$. Due to the symmetry of the random permutation,

$$\mathbb{E}\left[a_{i,t+1} I(r_{t+1} > \boldsymbol{a}_{t+1}^\top \boldsymbol{p}_{t+1})\right] \leq d_i + \frac{6\sqrt{2\bar{a}^2 m \log n}}{\sqrt{\min\{t, n-t\}}}.$$

Summing up the inequality, we have

$$\mathbb{E}[\boldsymbol{A}x - \boldsymbol{b}] \leq O(\sqrt{mn}\log n).$$

$\square$

## B7 Proof of Theorem 4

*Proof.* At time $t$, the optimal solution to the scaled LP is $\tilde{\boldsymbol{x}}^{(t)} = (\tilde{x}_1^{(t)}, ..., \tilde{x}_t^{(t)})$. We have

$$\mathbb{E}\left[r_t x_t\right] = \mathbb{E}\left[r_t \tilde{x}_t^{(t)}\right]$$

$$= \frac{1}{t}\mathbb{E}\left[\sum_{j=1}^{t} r_s \tilde{x}_j^{(t)}\right].$$

Then, for the objective,

$$R_n^* - \sum_{t=1}^{n}\mathbb{E}\left[r_t x_t\right] = R_n^* - \sum_{t=1}^{n}\frac{1}{t}\mathbb{E}\left[\sum_{j=1}^{t} r_s \tilde{x}_j^{(t)}\right]$$

$$\leq m\bar{r} + \frac{\bar{r}}{\underline{d}}\log n\sqrt{n} + m\bar{r}\log n + \bar{r}\max\{16\bar{a}^2, e^{16\bar{a}^2}, e\}.$$

where the second line comes from an application of Proposition 1. Then, we analyze the constraint violation. From the construction of the algorithm, we have that $\mathbb{E}[a_{it} x_t] \leq d_i$. Let

$$A_{it} = a_{it} x_t - d_i$$

and then we know

$$M_{it} = \sum_{j=n-t+1}^{n} A_{ij}$$

is a supermartingale with $|A_{ij}| \leq \bar{a} + \bar{d}$. Then if we apply Hoeffding's lemma for supermartingale, we have

$$\mathbb{P}\left(M_{in} \geq 2(\bar{a}+\bar{d})\sqrt{n}\log n\right) \leq \exp\left\{-\frac{2(\bar{a}+\bar{d})^2 n \log^2 n}{n(\bar{a}+\bar{d})^2}\right\}$$

$$\leq \exp\{-2\log^2 n\} \leq \frac{1}{n},$$

when $n > 3$. Thus,

$$\mathbb{E}\left[\left(\sum_{t=1}^{n} a_{it}x_t - d_i\right)^+\right] = \mathbb{E}\left[(M_{in})^+\right]$$

$$\leq 2(\bar{a}+\bar{d})\sqrt{n}\log n\, \mathbb{P}\left(M_{in} < 2(\bar{a}+\bar{d})\sqrt{n}\log n\right)$$
$$+ \bar{a}n\mathbb{P}\left(M_{in} \geq 2(\bar{a}+\bar{d})\sqrt{n}\log n\right)$$
$$\leq 2(\bar{a}+\bar{d})\sqrt{n}\log n + \bar{a}$$
$$\mathbb{E}\left[v(\boldsymbol{x})\right] \leq 2(\bar{a}+\bar{d})\sqrt{mn}\log n + \bar{a}\sqrt{m}.$$

$\square$

## C1    Algorithm Discussion

### Obtaining Feasible Solution

We present a simple approach to convert the solution obtained from Algorithm 1 into a feasible solution. Let $\boldsymbol{x} = (x_1, ..., x_n)$ be a solution by Algorithm 1, and $S_+ = \{t : x_t = 1, t = 1, ..., n\}$ be the index set of nonzero $x_t$'s and $n_+ = |S_+|$ be the cardinality of $S_+$. The idea is to randomly select a subset of $S_+$ and force $x_t = 0$ for indices in this subset. Note that the expected total constraint violation is sublinear in $n$, we only need to select a small proportion of $x_t$'s and force them to be 0. Specifically, define the maximum constraint violation quantity over all constraints:

$$v = \frac{1}{\sqrt{n}\log n}\max_{i=1,...,m}\left\{\left(\sum_{t=1}^{n} a_{it}x_t - b_i\right)^+\right\}.$$

Moreover, we require $v \geq 1$. We choose a set $S_0 \subset S_+$ uniformly with $|S_0| = \min\left\{\left\lceil \frac{2vn_+ \log n}{\underline{d}\sqrt{n}} \right\rceil + 1, n_+\right\}$, and let

$$\hat{x}_t = \begin{cases} 0, & t \in S_0 \\ x_t, & t \notin S_0 \end{cases}$$

for $t = 1, ..., n$. The following theorem characterizes the properties of $\hat{x}_t$.

**Theorem 5.** *If $n > \max\left\{16, \underline{d}^2, \left(\frac{6\bar{a}}{\underline{d}}\right)^4\right\}$ and $\sqrt{n} > \frac{12\bar{a}(\bar{r}+(\bar{a}+\underline{d})^2 m)\log n}{\underline{d}^2}$, then $\hat{\boldsymbol{x}} = (\hat{x}_1, ..., \hat{x}_n)$ is a feasible solution with probability at least $1 - \frac{2}{n}$. Also, a feasible solution $\boldsymbol{x}$ can be constructed based on $\hat{\boldsymbol{x}}$ s.t.,*

$$\mathbb{E}\left[R_n^* - \boldsymbol{r}^\top \boldsymbol{x}\right] \leq O((m + \log n)\sqrt{n})$$

*for all $m, n \in \mathbb{N}_+$. The results hold under both the stochastic input model and the random permutation model, and the expectation is taken with respect to $\mathcal{P}$ or the random permutation accordingly.*

Theorem 5 tells that in a large-$n$-small-$m$ regime, precisely when $n \geq O(m^2 \log n)$, we can easily obtain a feasible solution with high probability based on the output of Algorithm 1 by randomly selecting $O(\sqrt{n}\log n)$ number of $x_t$ and forcing them to be 0. Furthermore, the newly obtained solution does not change the regret much. The theorem provides a guideline of the implementation of Algorithm 1 for the binary LP setting when a feasible solution is desired.

### Feasible Online Algorithm

Algorithm 4 is another natural variant of Algorithm 1 that outputs feasible solutions. Compared with Algorithm 1, Algorithm 4 sets $x_t = 1$ only when the constraints permit. This design is more aligned with the online LP algorithms that guarantees feasibility. [20] provided a regret analysis framework for this type of feasible

algorithms, and the key is to analyze the stopping time of constraint violation and the remaining resources for binding constraints. In this paper, the assumptions on $(r_j, \boldsymbol{a}_j)$ are parsimonious and they might be not sufficient to derive an upper bound on these two key quantities. Numerically, we observe that this feasible algorithm, in comparison with Algorithm 1, does not compromise the performance in terms of the regret. We will elaborate more on its numerical performance in the next section and leave the regret analysis of this algorithm as an open question.

---

**Algorithm 4** Simple Feasible Algorithm

---

1: Input: $d$
2: Initialize $\boldsymbol{p}_1 = \boldsymbol{0}$
3: **for** $t = 1, ..., n$ **do**
4:     Set
$$\tilde{x}_t = \begin{cases} 1, & r_t > \boldsymbol{a}_t^\top \boldsymbol{p}_t \\ 0, & r_t \le \boldsymbol{a}_t^\top \boldsymbol{p}_t \end{cases}$$
5:     Compute
$$\boldsymbol{p}_{t+1} = \boldsymbol{p}_t + \gamma_t \left( \boldsymbol{a}_t \tilde{x}_t - \boldsymbol{d} \right)$$
$$\boldsymbol{p}_{t+1} = \boldsymbol{p}_{t+1} \vee \boldsymbol{0}$$
6:     If constraints permit, set $x_t = \tilde{x}_t$; otherwise set $x_t = 0$.
7: **end for**
8: Output: $\boldsymbol{x} = (x_1, ..., x_n)$

---

### Nonstationary Algorithm

We consider another variant of the algorithm that takes into account the resource consumption while doing the subgradient descent. The intuition is similar to the action-history-dependent algorithm in [20]. If excessive resources are consumed in the early periods, the remaining resource $\boldsymbol{b}_t$ will shrink, and this nonstationary algorithm will accordingly push up the dual price and be more inclined to reject an order. On the contrary, if we happen to reject a lot orders at the beginning and it results in too much remaining resources, the algorithm will lower down the dual price so as to accept more orders in the future. In numerical experiments, this nonstationary algorithm performs better, but it is still on the same order of regret and constraint violation as Algorithm 1. The open question is if there exists a first-order algorithm that is free of re-solving any linear programs and could achieve $O(\log n)$ regret, possibly under stronger statistical assumptions.

---

**Algorithm 5** Simple Nonstationary Algorithm

---

1: Input: $d$
2: Initialize $\boldsymbol{p}_1 = \boldsymbol{0}$, $\boldsymbol{b}_0 = \boldsymbol{b}$
3: **for** $t = 1, ..., n$ **do**
4:     Set
$$x_t = \begin{cases} 1, & r_t > \boldsymbol{a}_t^\top \boldsymbol{p}_t \\ 0, & r_t \le \boldsymbol{a}_t^\top \boldsymbol{p}_t \end{cases}$$
5:     Update
$$\boldsymbol{b}_t = \boldsymbol{b}_{t-1} - \boldsymbol{a}_t x_t$$
6:     Compute
$$\boldsymbol{p}_{t+1} = \boldsymbol{p}_t + \gamma_t \left( \boldsymbol{a}_t x_t - \frac{\boldsymbol{b}_t}{n - t} \right)$$
$$\boldsymbol{p}_{t+1} = \boldsymbol{p}_{t+1} \vee \boldsymbol{0}$$
7: **end for**
8: Output: $\boldsymbol{x} = (x_1, ..., x_n)$

---

### D1 Numerical Experiments

The first experiment compares the performance of Algorithm 1, Algorithm 4, and Algorithm 5 in terms of regret and constraint violation. Algorithm 1 is implemented with two different choices of step size $\gamma_t$. In

this experiment, $m = 10$, $a_{ij}$'s and $r_j$'s are sampled i.i.d. from Unif$[0, 2]$. For each value of $n$, we run 100 simulation trials and in each trial, $d_i$'s are sampled i.i.d. from Unif$[1/3, 2/3]$. The average regret and constraint violation over all the simulation trials are shown in Figure 1. We plotted normalized regret and constraint violation, which is absolute regret and constraint violation divided the optimal objective value and the $L_2$ norm of the constraint capacity, respectively. We observe that the step size of $1/\sqrt{n}$ results in larger constraint violation but smaller regret compared with the step size of $1/\sqrt{t}$. This is because for the step size of $1/\sqrt{n}$, the updating of the dual vector $\boldsymbol{p}_t$ is slower. Consequently, more requests will be accepted at early stage and the constraint violation is larger in the end. The non-stationary algorithm (Algorithm 5) performs better than the simple algorithm (Algorithm 1) with $\gamma_t = 1/\sqrt{t}$. Also, the feasible algorithm (Algorithm 4) guarantees feasibility, i.e. zero constraint violation; it produces slightly larger regret, but the regret is still on the order of $\sqrt{n}$.

(a) Regret

(b) Constraint Violation

Figure 1: Experiment with Uniform i.i.d. input

In the second experiment (Figure 2), we compare the three algorithms in a setting where the boundedness of the support of distribution $\mathcal{P}$ is violated. Specifically, $m = 10$, $a_{ij}$'s are generated i.i.d. from $\mathcal{N}(1, 1)$ and $r_j = \sum_{i=1}^{m} a_{ij} - \epsilon_j$ where $\epsilon_j \sim \text{Unif}(0, m)$. For each value of $n$, we run 100 simulation trials, and in each trial, $d_i$'s are sampled i.i.d. from Unif$[1/3, 2/3]$. In this experiment, the regret performances of Algorithm 1 (with step size of $1/\sqrt{t}$) and Algorithm 5 are quite close to each other, while Algorithm 5 still performs better in respect with constraint violation. The feasible algorithm (Algorithm 4) still achieves regret on the order of $\sqrt{n}$. Note that our theoretical results, also all the previous literature on online LP problem, rely on the boundedness assumption for the LP input. An open question is how to generalize these analyses to the case when the distribution $\mathcal{P}$ has an unbounded support.

(a) Regret

(b) Constraint Violation

Figure 2: Experiment with Gaussian i.i.d. input

The third experiment (Figure 3) presents a negative result on all three algorithms. Specifically, $a_{ij}$'s are generated i.i.d. from truncated Cauchy$(1, 1)$ (with different thresholds) and $r_j = \sum_{i=1}^{m} a_{ij} - \epsilon_j$ where $\epsilon_j \sim \text{Unif}(0, m)$. As before, for each value of $n$, we run 100 simulation trials, and in each trial, $d_i$'s are sampled i.i.d. from Unif$[1/3, 2/3]$. We observe that the performance becomes unstable as the truncation threshold goes up. The

phenomenon is consistent with the previous analysis that the algorithm regret is positively affected by the upper bound on $a_{ij}$'s and $r_j$'s. The empirical finding suggests that a light-tail distribution is probably necessary for an online LP algorithm to succeed.

(a) Regret

(b) Constraint Violation

Figure 3: Experiment with Cauchy i.i.d. input

Figure 4 presents the algorithm performance under the random permutation model. We first generate four groups of data with equal size from four different distributions and then combine these groups as the LP input: (i) $a_{ij}$'s are generated from Unif$[0, 2]$; (ii) $a_{ij}$ are generated from $\mathcal{N}(1, 1)$; (iii) $a_{ij}$ are generated from $\mathcal{N}(0, 1)$; (iv) $a_{ij}$ are generated from uniform distribution on $\{-1, 1, 3\}$. $r_j$'s for all four groups are generated from Unif$[0, 1]$. Note this data set can not be generated from any distribution in the stochastic input model. For each value of $n$, we run 100 simulation trials, and in each trial, $d_i$'s are sampled i.i.d. from Unif$[1/3, 2/3]$. In this experiment, we observe that the algorithms all achieves $O(\sqrt{n})$ regret except for Algorithm 1 with step size $1/\sqrt{n}$. The step size results in a negative regret but much larger constraint violation than the other algorithms. All the presented algorithms achieve $O(\sqrt{n})$ constraint violation.

(a) Regret

(b) Constraint Violation

Figure 4: Experiment with randomly permuted input

In addition, we conduct two groups of experiments to illustrate the computational aspect of our algorithms. The algorithms are implemented on a PC with Intel Core i7-9700K Processor. We use the Gurobi solver - one of the state-of-the-art LP and integer LP solvers - as benchmark and to compute the optimal solution. We emphasize that the code for our algorithms, unlike the Gurobi solver, is not fully optimized in the experiment, so there is still great room for improving the computational efficiency of our algorithms. Table 1 presents the experiments under the worst-case example for online LP problem in [3]. The example is constructed under the random permutation model, and it provides a lower bound on the right-hand-side of the constraint for the existence of an $(1 - \epsilon)$-competitiveness online LP algorithm. In this sense, it represents one of the most challenging problem instances under the random permutation model. Gurobi solves the binary LP problem with $1\%$ MIPGap and all competitiveness ratios in the table are reported against the optimal objective value of the relaxed LP's optimal objective. Gurobi computes the optimal solution in an offline fashion while the other three algorithms

are online. The numbers are computed based on an average over 100 different problem instances. Algorithm 1 is implemented with both step size of $\frac{1}{\sqrt{t}}$ and $\frac{1}{\sqrt{n}}$. We stop the algorithm and set the rest decision variables to be zero when any of the constraint is exhausted. Though Algorithm 2 and Algorithm 3 have provably smaller regret bounds, our algorithm provides better empirical performance. For small value of $m$, our algorithm outputs a near-optimal solution within much less time than Gurobi. Also, the competitiveness ratio decreases as $m$ goes larger, which is consistent with the $O(m)$ term in the regret bound of our algorithm.

|  |  | Gurobi | Alg. 1 ($\frac{1}{\sqrt{t}}$) | Alg. 1 ($\frac{1}{\sqrt{n}}$) | Alg. 2 | Alg. 3 |
|---|---|---|---|---|---|---|
| $m = 8, n = 10^3$ | CPU time | 0.082 | 0.023 | 0.015 | 126.684 | 126.696 |
|  | Cmpt. Ratio | 100.0% | 99.1% | 99.7% | 89.8% | 97.9% |
| $m = 128, n = 10^4$ | CPU time | 0.408 | 0.141 | 0.138 | 2338.149 | 2338.285 |
|  | Cmpt. Ratio | 100.0% | 99.3% | 98.8% | 94.8% | 97.7% |
| $m = 1024, n = 10^5$ | CPU time | 52.496 | 3.479 | 3.270 | >3000 | >3000 |
|  | Cmpt. Ratio | 100.0% | 94.6% | 98.7% |  |  |
| $m = 4096, n = 10^5$ | CPU time | 114.96 | 27.093 | 32.020 | >3000 | >3000 |
|  | Cmpt. Ratio | 99.6% | 73.1% | 83.3% |  |  |
| $m = 4096, n = 2 \times 10^5$ | CPU time | 254.243 | 57.541 | 53.887 | >3000 | >3000 |
|  | Cmpt. Ratio | 99.7% | 80.0% | 89.1% |  |  |

Table 1: Performance under worst-case example in [3]

In Table 2, we test the performance of our algorithm on some Multi-knapsack benchmark problems [7, 12]. As the last experiment, algorithm 1 is implemented with both step size of $\frac{1}{\sqrt{t}}$ and $\frac{1}{\sqrt{n}}$, and the experiment setup is the same as the last experiment. The competitiveness ratios are reported against the relaxed LP's optimal objective value. The computational advantage of our simple online algorithm is significant. Like the last experiment, although Algorithm 2 and Algorithm 3 reduces the regret upper bound by a factor of $\sqrt{m}$, the advantage of these two algorithms is not evident in practice. It deserves more efforts to understand whether the current regret bound for Algorithm 1 is tight. Also, it merits more study on how to design online algorithms that works effectively for a large-$m$-and-small-$n$ regime.

|  |  | Gurobi | Alg. 1 ($\frac{1}{\sqrt{t}}$) | Alg. 1 ($\frac{1}{\sqrt{n}}$) | Alg. 2 | Alg. 3 |
|---|---|---|---|---|---|---|
| $m = 5, n = 500$ | time | 0.116 | 0.006 | 0.006 | 70 | 70 |
|  | Cmpt. Ratio | 99.6% | 92.3% | 75.05% | 91.8% | 91.5% |
| $m = 10, n = 500$ | time | 0.136 | 0.006 | 0.006 | 132 | 132 |
|  | Cmpt. Ratio | 99.6% | 91.8% | 80.9% | 91.6% | 90.7% |
| $m = 30, n = 500$ | time | 95.2 | 0.006 | 0.005 | 134 | 133 |
|  | Cmpt. Ratio | 99.4% | 91.5% | 89.4% | 89.1% | 90.4% |
| $m = 10^3, n = 10^5$ | time | 857 | 2.711 | 2.679 | >3000 | >3000 |
|  | Cmpt. Ratio | 99.8% | 94.9% | 97.3% |  |  |
| $m = 3 \times 10^3, n = 10^5$ | time | 1069 | 16.63 | 16.61 | >3000 | >3000 |
|  | Cmpt. Ratio | 94.0% | 82.3% | 86.8% |  |  |
| $m = 3 \times 10^3, n = 2 \times 10^5$ | time | 2799 | 40.21 | 34.84 | >3000 | >3000 |
|  | Cmpt. Ratio | 93.8% | 84.7% | 88.4% |  |  |

Table 2: Multi-knapsack benchmark problem