[Reviews · NeurIPS 2020]

Review 1

Summary and Contributions: This paper deals with solving binary linear programs in an online form in the sense that the decision maker has to decide the t-th element of the solution at round t, in the beginning of which the t-th column and objective coefficient are revealed. The authors propose a simple algorithm based on dual descent and provide bounds for the (expected) difference from the optimal offline solution and constraint violation in two cases: iid columns and objective coefficient and them being selected by an adversary but revealed at a random order. They also analyze two algorithms with competitive ratio guarantees from the literature.

Strengths: The result and techniques used for deriving the bounds for the random permutation case, as well as the algorithms from the literature in Section 5 seem interesting.

Weaknesses: The setting examined has quite a lot of similarities with algorithms used in stochastic network optimization, see e.g. Neely, M.J., 2010. Stochastic network optimization with application to communication and queueing systems. Synthesis Lectures on Communication Networks, 3(1), pp.1-211. Therein, each constraint is assigned a "queue" , which tracks its violation and essentially tracks the optimal dual variables of the problem. Indeed, the regret bounds of Section 3 can be achieved by the following algorithm, which is a direct application of the aforementioned techniques: x_t = 1(V r_t -\sum_i Q^i_ta_t^i ) Q^i_{t+1} = [Q^i_t -d_i]^+ + a_t^i x_t, \forall i=1,2,..,m with V = 1/sqrt(n) It is not clear how big of a contribution is the generalization to negative coefficients in the constraint matrix.

Correctness: The claims and proofs seem correct.

Clarity: The paper is mostly well written, though the following parts are unclear: 1. "Our algorithm is inspired by the relaxed LP, and it directly outputs an integer solution to the relaxed LP.", lines 24-25 2. What exactly is meant by "artificially created randomness" in line 174 ? 3. "However, as pointed out by [10], one can always randomly perturb rt’s by arbitrarily small amount. In this way, the assumption will be satisfied, and the effect of this perturbation on the objective can be made arbitrarily small", lines 190-193.

Relation to Prior Work: Except the algorithm and the results of Section 3 (already mentioned in the "weaknesses part of the review), coverage of the literature seems adequate. It would be nice to mention what is the main technical difficulty behind online linear programs with negative coefficients (this work is apparently the first to address them).

Reproducibility: Yes

Additional Feedback: ----- Update ------- Thanks for clarifying the relation with the previous work and the unclear passages in the text.


Review 2

Summary and Contributions: - This paper proposes an algorithm for solving online linear and binary integer programs, with regret bounds under i.i.d. and permutation models. In this online model, the (r_j cost coefficient, a_t constraint column) are revealed sequentially. When the problem is an IP, an LP relaxation is performed and all performance guarantees are quantified via the LP. - A sublinear regret bound of O((m + log n) sqrt(n)) is proven for the optimality gap vs. the expected , as well as a constraint violation guarantee. The authors then prove that the proposed algorithms from [Agrawal '14, Kesselheim et al. '14] have better statistical performance, at a large computational cost.

Strengths: - The problem setting of online packing LPs is a classic and well-motivated one. - The algorithm is appealing in its simplicity. It is much more efficient than state-of-the-art algorithms from [Agrawal '14, Kesselheim et al. '14]. The tradeoff is quantified using regret bounds proven for those two results. - The usage of permutational Rademacher complexity to analyze these algorithms is interesting. Even though preceding work has not tended to appear in ML venues, in my opinion the analysis techniques place this paper within the scope of relevance for NeurIPS.

Weaknesses: - My primary concern is insufficient comparison with the existing literature on online LP, like the two works cited [Agrawal '14, Kesselheim et al. '14]: - The paper claims novelty in the sublihear competitive ratios obtained in those works of the form O(1 - \eps(m,n)), so that \eps(m,n) * OPT is the regret. From a glance at the works cited by [Agrawal '14], "Online stochastic packing applied to display ad allocation" [Feldman et al. '10] has an 1/OPT term in this competitive ratio, giving a sublinear regret bound. Some clarifying discussion is necessary here. - Moreover, the standard in the literature on this problem, starting with [Kleinberg '05], is to prefer dependences on B := min_i b_i (from the notation of [Agrawal '14], underbar-d in this paper) rather than OPT; see the related work section in that reference. Some discussion and a clearer comparison is necessary, since this line of work is so well-established. - It seems (at a glance; I haven't verified completely) that the positivity assumptions in those cited works, the removal of which is pointed out as a novel contribution in this work, comes not from some fundamental mathematical reason, but rather to simplify the sign conventions when the authors choose to quantify their results using competitive ratio on a positive utility function rather than regret. Some clarification about this would be appreciated; in any case, the manuscript should discuss this at greater depth. - It seems that the original algorithms in these cited works operate under the model where constraint violation is not permitted, while expected violation is considered as a cost here. Could the authors clarify on this discrepancy? - In summary, I like the ideas in the paper; however, since this line of work is so well-established and the problem is so concrete, this works needs to be more concrete and thorough in establishing its relationship with prior work.

Correctness: - The proofs appear to be correct.

Clarity: - The presentation is generally clear. - Small point on a confusing side remark: what do the authors mean when they say that the sqrt(m) factor gap is "the same as the lower and upper bound for OCO up to a log factor" (like 251)?

Relation to Prior Work: - The paper is thorough about mentioning related work, and does a particularly good job of bridging some disparate bodies of work. - However, there needs to be a more thorough clarification on the concrete technical differences between this work and the many others in this line of work; see above.

Reproducibility: Yes

Additional Feedback: *** post-response *** Thanks for your response. - If the "breaking constraints" section of the rebuttal checks out, it alleviates my concerns about that particular difference vs. prior literature, since the results then become directly comparable. For this reason, I'm increasing my score. - I wasn't aware until reading the other reviews that the algorithm is similar to the one seen in [Neely et al. '10]. However, there is enough novelty when considering the lines of work that are bridged. - I still believe a more in-depth discussion of the competitive ratio bounds of previous work (including the dependences on B) are in order.


Review 3

Summary and Contributions: The authors give a simple and fast algorithm for solving online binary integer linear programming and derive regret and violation bounds in both the stochastic model and random permutation model. The authors then go on to give regret and violation bounds for two previous, slower algorithms - which had, up until now, only competitive ratio bounds proven.

Strengths: Online integer linear programming is a very important and famous problem so I think any decent improvement here is an important result. Although the new algorithm has worse regret and violation bounds than the previous algorithms analysed in this paper its time complexity is only O(m) per trial - in terms of time complexity this is a significant improvement over (apparently) all other algorithms - that have to explicitly solve linear programs. The authors also give novel regret bound analyses of two previous online algorithms (whereas before we only had competitive-ratio bounds). Although, in my opinion, this result is not as significant as the new algorithm, it contributes to the strength of the paper.

Weaknesses: I personally think that breaking constraints is a weakness in an algorithm - however, the authors, in their response, gave a method for stopping the algorithm from breaking constraints with high probability.

Correctness: The paper appears correct, although I haven't read the proofs in the supplementary material so can't be sure.

Clarity: The paper is well written.

Relation to Prior Work: This is discussed and the work appears novel.

Reproducibility: Yes

Additional Feedback: After rebuttal: I have incorporated the author responses into the review. My score remains the same


Review 4

Summary and Contributions: The paper proposes a new online algorithm for solving integer linear programs and their LP relaxations. Compared to the existing online methods the algorithm has a significantly lower computational cost for a price of worse precision guarantees.

Strengths: The theoretical claims of the paper are sound and rigorously proved. I can not judge about novelty of the work since it lies on the periphery of my research interests, but - a quick search did not show me any sufficiently similar work - the referencing of the related works looks solid. Therefore, I assume the the proposed algorithm is sufficiently new.

Weaknesses: I. Experimental evaluation is too short and non-reproducible. 1) The experimental evaluation is in the supplement only. 2) The used benchmark datasets are not explicitly listed. Instead only references to 2 monographs are given. I quickly checked both and neither of these monographs contains instances of the benchmark problems. Therefore, it is unclear from the paper what problem instances where used. If they are generated then what are the generation rules. 3) The benchmark data seems to cover "some Multi-knapsack benchmark problems" only. The limited experimental evaluation does not cover the theoretical statements of the paper. The theory says the proposed method is fast, but less precise, the competing methods slow and more precise. In the experiments the proposed method is as precise as competitors or even better and much faster. Experiments could be more extensive to support the claims of the theory. II. Is NeurIPS a suitable venue for this paper? On one side yes, online optimization methods seems to belong to the focus of the NeuriPS community. On the other side, advantages of these works are usually demonstrated on machine learning applications if they are published at NeurIPS. This is not the case for this paper.

Correctness: The mathematical claims of the paper seems to be correct, at least I was unable to find any mistakes.

Clarity: Although being quite theoretical and mathy, it is very well written and easily readable given the corresponding background, since the level of details corresponds rather to a specialized optimization journal and not to the broader NeurIPS audience.

Relation to Prior Work: Yes. As mentioned above, I can not judge whether the list of related works is complete, however, the paper clearly articulates the differences to the previous contributions.

Reproducibility: Yes

Additional Feedback: Minor comments: An extensive number of special names for formulas, like ILP, P-LP, D-LP, SAA, s-S-LP, . Although this kind of referencing is popular in an optimization literature, it does not make reading easier compared to the case when a plain numbering is used. Especially, if the acronym for a formula is not obvious, like SAA. On the other side, I miss the numbering of other formulas, which I can not even reference neither in this review nor in a follow-up paper, if needed. l19: the sentences are confusing "we present ... a general class of integer linear programs (LP). Different specifications of the considered LP problem...", because it is unclear, whether you speak about LPs or ILPs. l34: an equivalent l37: "free of solving" is repeated twice. l78: "complementary condition" -> complementary slackness condition. To my taste, the derivation of the corresponding formula could contain somewhat more details. l 84: "of" the dual problem Algorithm 1: I would suggest to use the upperindex for iteration counter for "p", as the lower is used for the coordinates of x. ======== POST REBUTTAL====== Unfortunately, authors did not really address my main concern - the provided experimental evaluation. Therefore, I can not improve my score.

[Author Response · NeurIPS 2020]

We thank the reviewers for the careful feedback and appreciate the time spent reading our paper. Detailed responses are as below:

Literature on online LP (OLP) and the contribution of our work:

(i) From the algorithmic perspective, our algorithm has a strongly polynomial O(nnz(A)) flop complexity (linear in the number of non-zero entries in A), while the previous OLP algorithms all require solving O(log n) or O(n) of LPs (increasing to the full size over time). For example, Agrawal et al. (2014) solved O(log n) LPs and Kesselheim et al. (2014) solved O(n) LPs. As far as we know, the algorithm is the first of its kind and the most efficient OLP algorithm so far.

(ii) As to the analytical framework, we analyze the algorithm under both stochastic input model and random permutation model with minimal technical assumptions. As mentioned by the reviewer, our algorithms share similarity with the network control algorithm in Neely, M.J. (2010), but our analysis extends their analysis (in i.i.d. setting) to the random permutation setting. And the application of Permutational Rademacher Complexity is novel in online learning and regret analysis literature.

Technically speaking, we adopt a different method in analyzing the random permutation model than the works in OLP literature (such as Feldman et al., Agrawal et al., Kesselheim et al. etc.). They all applied a shrinkage technique and its idea is to perform the online learning as if the constraint is shrunk by a factor of $1-\varepsilon$, and then the output online solution would be feasible with high probability for the original problem. We provide an alternative treatment in the paper by direct relating the past and future observations (through Permutational Rademacher complexity and other related derivations). Then it enables us to (i) analyze a non-shrinkage version of Agrawal et al. (2014) and (ii) relax the previous non-negativeness assumption.

Breaking constraints: In the updated version of the paper, we will provide an algorithm that is feasible with high probability and the regret of the new algorithm absorbs the current constraint violation into the current regret. The idea is very straightforward: We could randomly select $O(\sqrt{n})$ of decision variables based on the online solution obtained by the current algorithm and switch them from one to zero. Intuitively, the switching operation will reduce the total constraint consumption so that the new solution is provably feasible based on a concentration argument. In this sense, the bi-objective performance measure in our paper is more for a better exposition but not entailed by our analytical framework.

Besides, the notion of "Artificially created randomness" refers to the application of our algorithm in solving integer LPs. The original inputs of an integer LP might not satisfy the i.i.d. assumption. When this is the case, we can random permute the inputs of the integer LP to make it satisfy the random permutation assumption, and then our algorithm and analysis can be applied. In this way, the randomness can be artificially created when it is not inherent. We also appreciate the typos pointed out by the reviewers. We will correct these typos and include more numerical experiments in the updated paper. We will highlight (i) the effectiveness in large-scale problems and (ii) our fast solution as a warm start for other exact LP solvers.

[Meta-Review · NeurIPS 2020]

The paper concerns online (approximate) solving of a general class of binary linear programs. The analysis of the algorithm is conducted under two models: stochastic inputs (columns of LP drawn i.i.d.) and random permutation model (columns of LP revealed in a random order). The authors develop a simple and fast online algorithm performing stochastic subgradient descent on a dual problem with provable guarantees on the regret and constraint violation. The paper received borderline reviews with a slight lean towards acceptance. The main strengths of the paper are: - New techniques for deriving the regret bounds in the random permutation model (permutational Rademacher complexity). - Appealing simplicity and efficiency of the algorithm: its time complexity is only O(m) per trial (m = number of rows of the constraint matrix A), while all other algorithms require explicitly solving linear programs. - Novel regret bounds of two previous algorithms (from past work) in the random permutation model. The main weaknesses were: - Insufficient comparison with the existing online LP literature, in particular with the competitive ratio bounds of previous work. - A similarity with the algorithm by Neely (2010). - The removal of positivity assumptions from the previous work, pointed out as a novel contribution, might not be technically difficult. The reviewers also pointed out some issues with the presentation which need to be fixed, and that the experimental evaluation is too short and non-reproducible. I urge the authors to take into account all the remarks of the reviewers in the final version of the paper, in particular: a clarified discussion on the competitive ratio bounds in the previous work, technical significance of handling negative signs in the constraint matrix, absorbing constraint violation in the regret, and similarity to the algorithm by Neely (2010).